# 5-HT$_2$ antagonism indirectly increases motor unit discharge rate and cervicomedullary motor evoked potential amplitude during submaximal elbow flexions

Tyler T. Henderson[1,2], Janet L. Taylor[3], Jacob R. Thorstensen[4] and Justin J. Kavanagh[2]

[1] *Faculty of Health, Charles Darwin University, Darwin, Australia*
[2] *Griffith Health, Griffith University, Gold Coast, Australia*
[3] *Neuroscience Research Australia, Sydney, Australia*
[4] *Faculty of Health Sciences and Medicine, Bond University, Gold Coast, Australia*

Handling Editors: Richard Carson & Jakob Škarabot

The peer review history is available in the Supporting information section of this article (https://doi.org/10.1113/JP288317#support-information-section).

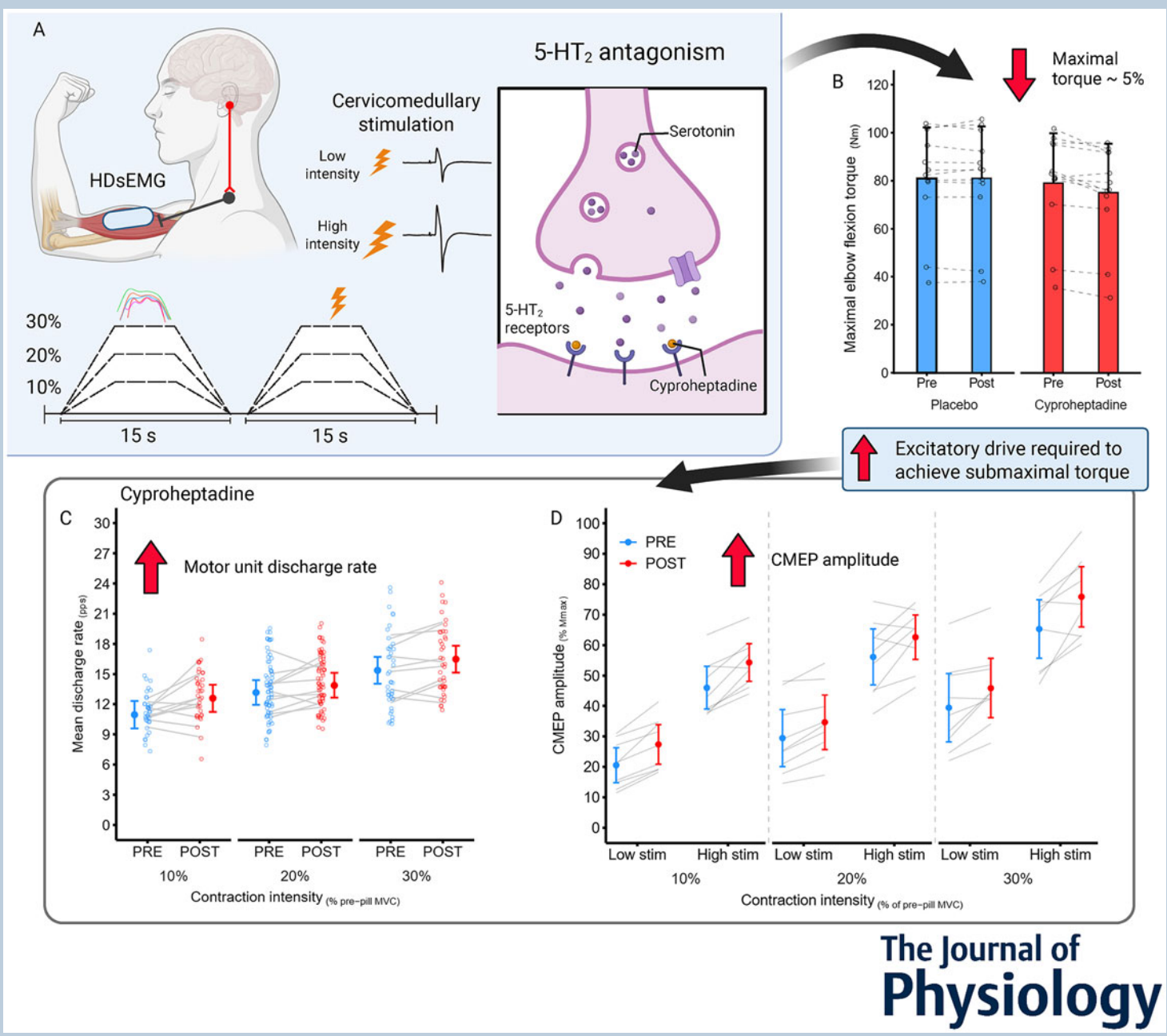


**Abstract figure legend** Motoneurone excitability of the elbow flexors was assessed in healthy human participants before and after the ingestion of the 5-HT$_2$ receptor antagonist cyproheptadine or a placebo. *A*, participants were seated upright, with their right arm attached to a custom torque transducer with bipolar EMG and a 64-channel high-density EMG array positioned over the right biceps brachii muscle. Low and high intensity stimulation was delivered to the cervicomedullary region to examine excitability of different proportions of the motoneurone pool. *B*, cyproheptadine was found to decrease maximal torque by ~5%. *C* and *D*, during submaximal contractions, both motor unit discharge rates (*C*) and cervicomedullary motor evoked potential (CMEP) amplitudes (*D*) from low and high intensity stimulation increased with 5-HT$_2$ antagonism. These findings suggest that additional voluntary drive is required to achieve the same torque level with 5-HT$_2$ receptor antagonism, indirectly enhancing motoneurone activity and excitability of the motoneurone pool.

Ethical approval statement: Ethical approval was obtained via the Human Research Ethics committee at Griffith University (Griffith University Reference Number: 2023/153), and written informed consent was obtained from all participants prior to data collection.

**Abstract** Spinal motoneurone excitability is heavily regulated by serotonin via somatodendritic $5\text{-}HT_2$ receptors. However, the effects of these receptors on the excitability of motoneurones in the upper limb are not clearly understood. Therefore, the purpose of this study was to assess the effects of $5\text{-}HT_2$ antagonism on motor unit discharge characteristics of the biceps brachii and evoked responses to cervicomedullary stimulation. Twelve healthy individuals (aged $24 \pm 3$ years) participated in this double-blind, placebo-controlled, two-way crossover trial and were administered the $5\text{-}HT_2$ antagonist cyproheptadine. high-density surface EMG (HDsEMG) was used to examine motor unit activity in the biceps brachii during trapezoidal contractions of 10%, 20% and 30% maximal voluntary contraction (MVC). Cervicomedullary stimulation was used to produce small and large cervicomedullary motor evoked potentials (CMEPs) in the elbow flexors during these sub-maximal contractions. Cyproheptadine reduced maximal elbow flexion torque ($\sim$5%, $P = 0.003$), and increased EMG amplitude ($\sim$2%, $P = 0.037$), motor unit discharge rates ($\sim$1.5 pulses/s, $P = 0.001$) and CMEP amplitude from low ($\sim$25%, $P = 0.002$) and high ($\sim$15%, $P = 0.026$) intensity stimulation during submaximal contractions. This is the first study to examine the effects of $5\text{-}HT_2$ antagonism on motoneurone excitability using both HDsEMG and cervicomedullary stimulation in a single experiment. The results of this study provide novel evidence that $5\text{-}HT_2$ receptor antagonism increases both motor unit discharge rates and CMEP amplitude during elbow flexions when torque targets remain unchanged from baseline.

(Received 8 December 2024; accepted after revision 17 June 2025; first published online 12 July 2025)

**Corresponding author** T. T. Henderson: Faculty of Health, Charles Darwin University, Casuarina campus, Northen Territory, Australia, 0810.    Email: tyler.henderson@cdu.edu.au

**Key points**

- Animal models have revealed that serotonin can heavily regulate motoneurone gain via $5\text{-}HT_2$ receptors.
- Recently, it has been revealed that $5\text{-}HT_2$ receptors can modulate motor unit firing characteristics in humans. However, the effect of these receptors on motoneurone excitability of the upper limb is not clearly understood.
- This study paired high-density surface EMG with cervicomedullary stimulation to provide a novel insight to the effects of $5\text{-}HT_2$ antagonism on motoneurone excitability.
- $5\text{-}HT_2$ antagonism reduced maximal elbow flexor torque, and increased motor unit discharge rates and cervicomedullary motor evoked potential amplitudes during submaximal contractions.
- These findings suggest that compensatory voluntary drive is required to achieve the same torque level with $5\text{-}HT_2$ receptor antagonism, indirectly enhancing motoneurone activity and excitability of the motoneurone pool.

## Introduction

Voluntary muscle activation is heavily regulated by neuro-modulatory inputs from the brainstem locus coeruleus and raphe nuclei, whereby noradrenergic (NA) and serotonergic (5-HT) systems modulate cortical and spinal motor circuits. In particular, the serotonergic raphe-spinal pathway has substantial impact on spinal motoneurone

**Tyler Henderson** recently completed his PhD at Griffith University, Gold Coast, Australia. His research has explored the role of serotonin in human motor activity through the combination of pharmacological interventions with non-invasive stimulation techniques and electromyography. His research interests focus on the brainstem contributions to muscle activation and, more broadly, the nervous system's regulation of motor control in humans.

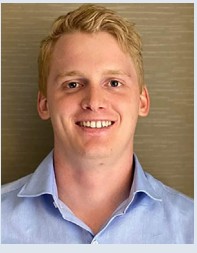

firing via activation of 5-HT$_2$ and 5-HT$_{1A}$ receptors to facilitate or inhibit motor activity, respectively (Cotel et al., 2013; Heckman et al., 2008; Kavanagh & Taylor, 2022; Perrier & Cotel, 2008; Perrier & Delgado-Lezama, 2005; Perrier et al., 2017). Cellular and animal models have revealed that facilitation of motor activity is mediated by 5-HT$_2$ receptors, which promote dendritic persistent inward currents (PICs) to facilitate motoneuronal excitability and self-sustained firing (Binder et al., 2020; Hounsgaard et al., 1988; Perrier & Hounsgaard, 2003). Recently, an abundance of evidence has confirmed these serotonergic effects in humans by using pharmacological interventions to manipulate 5-HT release and activation of 5-HT receptors within the CNS. Competitive antagonism of 5-HT$_2$ receptors has been found to modulate cortical and spinal motoneuronal excitability (Henderson et al., 2023; Thorstensen et al., 2021, 2022), as well as motor unit firing characteristics (Goodlich et al., 2023, 2024) in separate studies. Hence, the ingestion of a 5-HT$_2$ antagonist can not only reduce motor unit firing rates and estimates of PICs in the tibialis anterior (Goodlich et al., 2023), but also modulate motoneurone excitability and reduce maximal elbow flexion torque production by $\sim$5% (Henderson et al., 2023; Thorstensen et al., 2021, 2022).

Despite the known effects of 5-HT$_2$ receptor antagonism on motoneurone activity, we recently reported a contrasting result, as 5-HT$_2$ antagonism was found to increase cervicomedullary motor evoked potential (CMEP) amplitude during voluntary elbow flexions but not at rest (Henderson et al., 2023). We postulated that CMEP amplitude increased with 5-HT$_2$ antagonism due to an increased proportion of motoneurones close to firing threshold (i.e. in the subliminal fringe), which allowed more motoneurones to be more readily recruited into the CMEP. Under conditions of 5-HT$_2$ antagonism, we rationalised that the number of motoneurones in the subliminal fringe was increased due to the necessary increase in descending drive to motoneurones to produce the same level of torque (Henderson et al., 2023). The underlying mechanism for this is probably associated with reduced PIC amplitude when 5-HT$_2$ receptors are antagonised (Goodlich et al., 2023). With reduced PICs in active motoneurones, intrinsic sources of depolarisation are reduced, whereby greater synaptic excitation is required to cause these neurones to fire fast enough to produce the desired torque. Thus, greater voluntary drive is needed for increased excitation of active motoneurones, and this increased descending drive would spill over onto inactive motoneurones increasing their likelihood to discharge. Additionally, reduced PIC amplitude would increase the membrane resistance of the active motoneurones, making the active motoneurone pool more susceptible to discharge with additional synaptic input provided by cervicomedullary stimulation. Hence, the increase in CMEP amplitude is likely attributable to an 'indirect' effect of drug ingestion whereby increased excitatory drive is required to achieve the same level of torque due to altered motoneurone excitability caused by 5-HT$_2$ antagonism. Given that 5-HT$_2$ receptors can facilitate motoneurone discharge rates via promotion of PICs, and lower threshold motoneurones are more dependent on PICs than larger threshold motoneurones (Johnson et al., 2017; Lapole et al., 2023; Mesquita et al., 2020; Revill & Fuglevand, 2011), it is likely that the magnitude of effect of 5-HT$_2$ antagonism is different across the motoneurone pool.

The primary purpose of this study was to examine the effects of 5-HT$_2$ antagonism on biceps brachii motoneurone excitability using high-density surface EMG (HDsEMG) and cervicomedullary stimulation during voluntary elbow flexions. A secondary purpose of this study was to assess the effects of 5-HT$_2$ antagonism on different proportions of the motoneurone pool. HDsEMG was used to examine motor unit firing characteristics during elbow flexions to 10, 20 and 30% of peak torque. Low intensity cervicomedullary stimulation was used to produce smaller CMEP responses, and high intensity cervicomedullary stimulation was used to produce larger CMEPs. These two stimulation intensities were used to probe smaller and larger proportions of the motoneurone pool made up of mostly lower threshold motoneurones, and both lower and higher threshold motoneurones, respectively. Cervicomedullary stimulations were delivered during elbow flexions to 10, 20 and 30% of peak torque. CMEP amplitudes from low and high intensity stimulation, accompanied by non-invasive measures of motor unit spiking activity, can provide novel insight into the serotonergic modulation of motoneurones with different biophysical properties. We hypothesised that peak elbow flexion torque would be reduced, and CMEPs produced by low and high intensity stimulation would increase following ingestion of the 5-HT$_2$ antagonist cyproheptadine. We also hypothesised that motor unit discharge rates would decrease with 5-HT$_2$ antagonism.

## Methods

### Experimental design

This study employed a human, double-blind, placebo-controlled, two-way cross-over design. Participants were required to attend the laboratory on two separate occasions where a placebo or cyproheptadine capsule was ingested. HDsEMG and CMEP responses were obtained pre-pill ingestion (baseline) and post-pill ingestion for both placebo and cyproheptadine conditions. Testing sessions were separated by at least

7 days to ensure residual drug effects did not influence the second testing session (Paton & Webster, 1985).

## Participants and ethical approval

Prior to participant recruitment, a sample size calculation was completed to estimate the number of required participants to achieve statistical power. An effect size (0.45) observed from measures of evoked potentials under similar conditions in our previous work (Henderson et al., 2023; Thorstensen et al., 2022), an $\alpha$-level of 0.05 and power of 0.8 was used in this calculation, where an estimated 10 participants was required for this experimental design. Considering possible experimental challenges (e.g. discomfort from electrical stimulation) and potential participant drop-out, 14 healthy individuals were recruited to participate in this study. Each participant was screened using a medical history questionnaire with exclusion criteria specific to cyproheptadine contra-indications, electrical stimulation, and musculoskeletal injury. Individuals who were free from neurological impairment, were not taking any CNS acting medications (including antidepressants), were not pregnant, were without implanted metal objects (including pacemakers), and had no recent history of head, neck or upper limb musculoskeletal injuries were included in this study. Participants were required to attend a familiarisation session to ensure that CMEP responses could be produced from cervicomedullary stimulation. Following screening, 12 individuals (24 $\pm$ 3 years, two females, height: 178 $\pm$ 8 cm, weight: 80 $\pm$ 10 kg) participated in the study. Participants were required to refrain from any stimulants or depressants, and fatiguing exercise for at least 18 hours prior to each testing session. Ethical approval was obtained for this study from the Griffith University Human Research Ethics committee (GU reference number: 2023/153), and all participants provided written informed consent prior to any testing procedures. All testing procedures conformed to the standards set by the *Declaration of Helsinki* except for registration in a database.

## Drug intervention

A single 8 mg oral dose of the 5-HT$_2$ receptor antagonist cyproheptadine, and a placebo that consisted of Avicel filler, were compounded to opaque capsules. Capsules were ingested by participants on separate testing days where the order of placebo and drug administration was counterbalanced to avoid order effects. Cyproheptadine is a competitive antagonist of the 5-HT$_2$ receptor and has a high binding affinity for 5-HT$_{2A}$, 5-HT$_{2B}$ and 5-HT$_{2C}$ receptor subtypes (Honrubia et al., 1997). Post-pill testing began 2.5 h after drug administration to coincide with

previously reported drug effects that cyproheptadine has on neurophysiological measures (Goodlich et al., 2024; Henderson et al., 2023; Thorstensen et al., 2022; Wei et al., 2014).

## Participant set-up and electromyography

Participants were seated upright in a customised Biodex chair (Shirley, NY, USA) with the shoulder and elbow of their right arm flexed at 90 degrees (Fig. 1), and their wrist secured firmly to a custom-built transducer designed to measure elbow flexion torque (200 kg capacity, PT4000 S-Type, PT Ltd., Aukland, New Zealand). A computer monitor was positioned at eye level $\sim$1 m directly in front of participants to provide real time feedback of torque production. Bipolar surface EMG signals were obtained from the right biceps brachii via Ag/AgCl electrodes (Kendall ARBO, 24 mm diameter, CardinalHealth, Sydney, Australia) to assess evoked responses to cervicomedullary and brachial plexus stimulation. One electrode was placed slightly lateral ($\sim$2 cm) to the mid-point of the muscle belly and one electrode placed over the distal tendon of the biceps brachii ($\sim$4 cm interelectrode distance). These locations were identified by palpation by an experienced investigator. Bipolar EMG was amplified ($\times$100), and bandpass filtered between 10 and 1000 Hz using a second order Butterworth filter (CED 1902, Cambridge Electronic Design Ltd, Cambridge, UK). Torque and bipolar EMG data were sampled at 2000 Hz using a Power 1401 data acquisition interface with Signal software

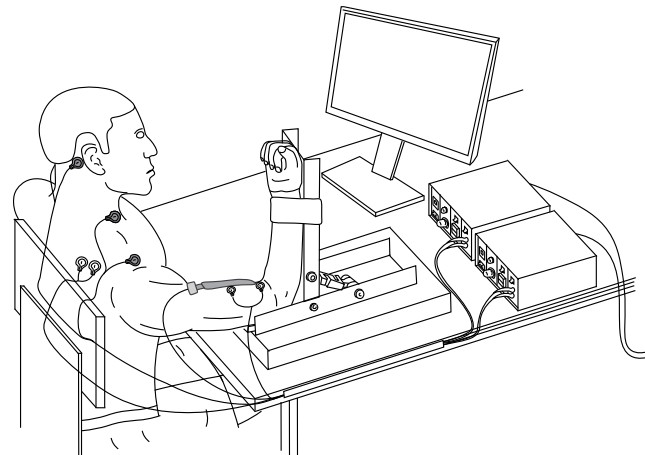

**Figure 1. Participant set-up**
Participants were seated upright, with their right arm attached to a custom torque transducer. Elbow flexion torque was measured with the elbow and shoulder joints flexed at 90 degrees. Bipolar EMG and a 64 channel (five columns by 13 rows) high-density EMG array were positioned over the biceps brachii muscle. Electrodes for electrical stimulation were positioned over the brachial plexus (Erb's point) and slightly inferior and medial to the left and right mastoid process.

(version 7, Cambridge Electronic Design). To assess motor unit activity of the biceps brachii, a 64-channel HDsEMG grid electrode (13 rows × 5 columns) with an interelectrode distance of 8 mm (OTBiolettronica, Torino, Italy) was positioned over the bulk of the biceps brachii muscle belly where the lateral edge did not overlap the proximal bipolar EMG electrode. Monopolar HDsEMG signals were amplified (×256), processed and sampled at 2000 Hz using a wireless amplifier (16-bit, Sessantaquattro, OTBioelettronica), and visualised using OTBioLab+ software (v.1.3.0, OTBiolettronica).

### Brachial plexus stimulation

Maximal compound muscle action potentials ($M_{MAX}$) were evoked in the biceps brachii using a constant current stimulator (0.2 ms pulse width, DS7AH, Digitimer Ltd, Welwyn Garden City, UK) and Ag/AgCl electrodes (Kendall ARBO, 24 mm diameter). A surface cathode was positioned over the right supraclavicular fossa (Erb's point) and a surface anode was positioned on the right acromion process to deliver single electrical stimuli to nerve fibres of the brachial plexus. To determine $M_{MAX}$ amplitude, the stimulation intensity was gradually increased in increments of 5–20 mA until a clear plateau was identified in the M-wave amplitude. Stimulator intensity was set at 130% of the intensity required to produce $M_{MAX}$ in the EMG signal of the resting biceps brachii. Stimulator intensity (134 ± 4 mA) remained unchanged throughout each testing session and was not adjusted after pill ingestion. To confirm reliability of M-wave measures across testing sessions, intraclass correlation coefficients (ICC) were calculated for $M_{MAX}$ amplitude between pre-pill and post-pill sessions with two-way mixed effects models using the psych package, version 2.5.3 (Revelle, 2025), in R Studio. These models revealed excellent reliability between sessions in the placebo condition (10% maximal voluntary contraction (MVC): ICC = 0.98, 95% CI = 0.93, 0.99; 20% MVC: ICC = 0.98, 95% CI = 0.90, 0.99; and 30% MVC: ICC = 0.97, 95% CI = 0.87, 0.99) and the cyproheptadine condition (10% MVC: ICC = 0.95, 95% CI = 0.83, 0.99; 20% MVC: ICC = 0.96, 95% CI = 0.82, 0.99; and 30% MVC: ICC = 0.97, 95% CI = 0.86, 0.99).

### Cervicomedullary stimulation

A second constant current stimulator (0.2 ms pulse width, DS7AH, Digitimer) was used to deliver single electrical stimuli at the cervicomedullary junction to elicit cervicomedullary motor evoked potentials (CMEPs) in the right elbow flexors. A surface cathode and anode were positioned slightly inferior (~1 cm) and medial to the left and right mastoid process. Two forms of cervicomedullary stimulation were used in this study. The 'low' stimulus intensity was set to produce a CMEP ~20% of $M_{MAX}$ peak-to-peak amplitude during contractions of 10% MVC. This intensity was used to recruit a smaller portion of the motoneurone pool predominantly made up of active motoneurones, or motoneurones close to firing threshold. A second stimulus intensity was set to produce a CMEP ~50% of $M_{MAX}$ peak-to-peak amplitude during contractions of 10% MVC. This 'high' intensity cervicomedullary stimulation would recruit a larger portion of the motoneurone pool including both lower- and higher-threshold motoneurones, particularly during high intensity contractions. These stimulator intensities remained fixed throughout each testing session and were not adjusted after pill ingestion (low intensity: 140 ± 20 mA, high intensity: 175 ± 26 mA). The onset latency of CMEP responses (~8 ms) was closely monitored to ensure corticospinal, and not cervical axons, were activated (Taylor, 2006; Taylor & Gandevia, 2004). Following screening protocols, CMEP responses were collected for nine participants due to shifts in onset latency and high levels of discomfort.

### Experimental protocol

**Calibration contractions.** Participants firstly performed four brief (~4 s) MVCs of the elbow flexors (Fig. 2). The largest torque produced by these contractions was used as the participant's peak torque, and all submaximal torque targets were based on this measurement. Participants then performed brief (~5 s) contractions for the calibration of motor unit filters (for offline HDsEMG analysis) and setting cervicomedullary stimulation intensities (see 'Cervicomedullary stimulation' above). For motor unit filter calibration, three brief contractions (5 s) were performed at each intensity of 15, 25 and 35% of peak torque. The additional calibration contractions of 15, 25 and 35% were used to train the motor unit filters at six different torque levels (10, 15, 20, 25, 30 and 35%) rather than three torque targets. The reason for this was to improve the sensitivity of motor unit decomposition across a wider range of recruitment thresholds. A secondary purpose of this was to increase the likelihood of identifying additional higher threshold units that may be recruited with cyproheptadine as a result of increased descending drive. Rest periods of 1–4 min were provided between contractions to mitigate the effects of neuromuscular fatigue.

**Ramped contraction protocol.** Once calibration contractions were completed, participants performed ramped (trapezoidal) contractions to 10, 20 and 30% of peak torque (Fig. 2). Each contraction was 15 s in duration, with a 5 s ramp-up phase, 5 s plateau and a 5 s ramp-down

phase. For motor unit analyses, four trials of these ramped contractions were performed in blocks at each intensity. For stimulation trials, these same ramped contractions were performed, with a single stimulation delivered in the middle of the plateau phase of the contraction. Five trials of each contraction intensity were performed for brachial plexus stimulation, and 10 trials at each contraction intensity were completed for both low and high intensity cervicomedullary stimulation in blocks of the same contraction intensity. The order of contraction intensity for HDsEMG contractions and stimulation contractions were randomised for each participant. Contractions were performed with ∼30 s rest between trials, and 1–3 min rest between blocks of contractions. Following the completion of the calibration, ramped contractions, and stimulation trials, participants ingested a placebo or cyproheptadine capsule. About 2.5 h following drug ingestion, participants were required to perform the same maximal contractions, submaximal ramped contractions and stimulation trials as pre-pill ingestion. Despite any changes to post-pill peak torque, submaximal torque targets remained the same as pre-pill ingestion. Pre- and post-pill testing sessions were ∼2 h in duration.

## Data analysis

**MVC torque and evoked potentials.** Analysis of peak elbow flexion torque, root mean square (RMS) EMG amplitude and evoked responses ($M_{MAX}$ and CMEPs) were performed offline using Signal software (version 7, Cambridge Electronic Design). RMS EMG amplitude was measured as RMS of the biceps brachii bipolar EMG signal. RMS EMG was assessed over a 400 ms period to encompass peak torque during maximal elbow flexions and immediately prior to cervicomedullary stimulation during submaximal contractions. $M_{MAX}$ and CMEP responses were measured as the peak-to-peak amplitude of the biphasic waveform for each contraction. This was conducted by setting a horizontal cursor at 0 mV, and the evoked responses were measured from the first clear deflection of the EMG signal following the stimulus artefact, to where the EMG signal returns to the 0 mV line following both phases of the waveform. An average peak-to-peak amplitude of all trials for each evoked potential at each intensity and contraction strength was calculated. Average CMEP amplitude at each contraction and stimulation intensity was then normalised to the average $M_{MAX}$ amplitude at the same contraction intensities within each testing session of each testing day. The percentage change ($\Delta$) from pre-pill (baseline) to post-pill sessions was calculated for evoked potentials using the formula: $\Delta = (\frac{\text{post pill} - \text{pre pill}}{\text{pre pill}}) \times 100$, so that a positive change would indicate an increase in that particular variable following pill ingestion. These change scores allowed for the direct comparison of change caused by placebo and cyproheptadine ingestion.

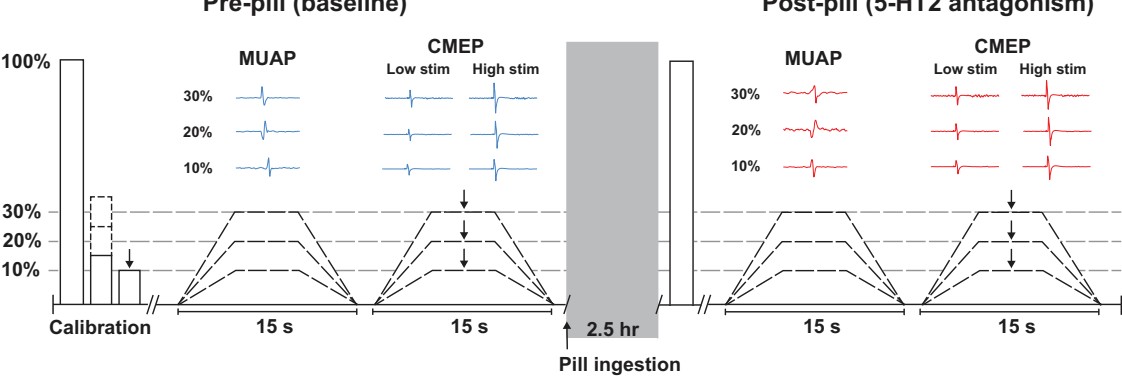

**Figure 2. Experimental protocol**
Participants performed elbow flexion tasks pre- (baseline) and post-pill ingestion of the 5-HT$_2$ receptor antagonist cyproheptadine or a placebo. High-density surface electromyography (HDsEMG) recordings were obtained from a 13×5 grid electrode overlaying the biceps brachii, and cervicomedullary stimulation was delivered at low (low stim) and high (high stim) stimulation intensities to produce cervicomedullary motor evoked potentials (CMEPs) in the elbow flexors. Participants performed maximal elbow flexions to establish submaximal torque targets. Following these maximal contractions, calibration contractions were performed to establish motor unit filters and stimulation intensities. Participants then performed ramped contractions to 10%, 20% and 30% of their maximal torque. These contractions were 15 s in duration, with a 5 s ramp up, 5 s plateau and 5 s ramp down phase. Following four successful trials at each contraction intensity, participants performed these same contraction tasks with cervicomedullary stimulation delivered in the middle of the plateau phase. Ten trials were completed at each contraction intensity for each cervicomedullary stimulation intensity of low stim and high stim. Following baseline testing, participants ingested either a placebo or cyproheptadine (8 mg) capsule. 2.5 h post-pill ingestion, the same contraction tasks were completed as baseline, with submaximal torque targets remaining the same level as pre-pill. MUAP, motor unit action potential. [Colour figure can be viewed at wileyonlinelibrary.com]

In addition to providing a placebo control for the drug condition, these change scores were used for the direct comparisons of change between low intensity and high intensity stimulation within each condition. Such comparisons were used to provide insight to serotonergic effects on smaller proportions and larger proportions of the motoneurone pool. To confirm that evoked potentials were indeed obtained from the biceps brachii muscle, $M_{MAX}$ and CMEP amplitudes from a sub-set of participants were also analysed using a bipolar configuration from the HDsEMG electrode recordings, whereby similar physiological responses were obtained from both forms of bipolar EMG analyses and yielded consistent results. For example, in one contraction intensity, peak-to-peak amplitude was ∼74% of $M_{MAX}$ for the bipolar configuration and ∼70% of $M_{MAX}$ for the HDsEMG configuration, with a relative change between sessions of −1.2 and −1% respectively. Maximal torque and evoked responses are presented as means ± SD in figures and tables unless otherwise stated.

**Motor unit identification and tracking.** Recorded HDsEMG files were processed offline using the DEMUSE tool (University of Maribor, Slovenia). A second-order Butterworth filter (20–500 Hz) was applied to the HDsEMG signals prior to decomposition using blind source separation with convolution kernal compensation (Holobar & Zazula, 2007). Fifty decompositions were completed independently for each recorded contraction (i.e. 10, 20 and 30% ramped contractions pre- and post-pill ingestion). Once decomposed, individual motor unit filters were applied to concatenated signals of pre- and post-pill ingestion files for respective contraction intensities and testing days (i.e. 10% pre-placebo and 10% post-placebo). These filters identified and tracked the same motor unit action potential (MUAP) shape across pre- and post-pill testing sessions to assess any changes to motor unit firing characteristics of the same motor unit following drug ingestion. To ensure the accuracy of the identification and tracking of motor units, manual inspection of each individual unit was completed, where all duplicate units identified within the same contraction intensity were removed from the analysis, and a minimum pulse to noise ratio of 30 dB was set for inclusion criteria in the analysis. Motor units were not tracked across days nor contraction intensities.

**Calculation of motor unit firing characteristics.** Firing characteristics were calculated from instantaneous estimates of discharge for tracked units from pre-pill to post-pill ingestion for placebo and cyproheptadine testing sessions. Mean discharge rate was calculated as the average discharge rate for the individual motor units during the plateau phase of the trapezoidal contractions. The percentage change (Δ) from pre-pill (baseline) to post-pill sessions was calculated for motor unit firing rates using the formula: $\Delta = (\frac{\text{post pill} - \text{pre pill}}{\text{pre pill}}) \times 100$. A positive change score would indicate an increase in firing rate from baseline following drug or placebo ingestion. Recruitment and derecruitment thresholds were calculated as the torque produced at the onset and offset of firing for the respective motor unit.

### Statistical analysis

Normality of data was determined using Shapiro–Wilk tests, and sphericity was assessed using Mauchly's test of sphericity for all measures. If data were non-spherical, Greenhouse–Geisser corrections were applied. All statistical tests were performed in R Studio (R software, version 4.3.1). Statistical significance was set at $P < 0.05$ for all statistical analyses. For peak elbow flexion torque, a two-way repeated measures ANOVA was used to assess a main effect of drug condition (placebo and cyproheptadine), testing session (baseline and post-pill) and any interaction effect of drug condition by testing session for maximal torque production.

To examine whether baseline evoked potentials were similar across drug conditions (placebo and cyproheptadine), one-way repeated measures ANOVA was separately applied to evoked potential amplitudes ($M_{MAX}$, low intensity CMEP and high intensity CMEP). Data were then pooled from baseline placebo and baseline cyproheptadine sessions to assess the effects of both contraction intensity and stimulation intensity on CMEP amplitudes. To do this, a two-way repeated measures ANOVA was applied to pooled baseline CMEP responses to examine a main effect of contraction intensity (10%, 20% and 30%), stimulation intensity (low and high intensity stimulation) and an interaction effect of contraction intensity by stimulation intensity. To assess the effects of placebo and cyproheptadine ingestion on CMEP amplitude (i.e. pre to post comparisons for placebo cyproheptadine conditions), separate two-way repeated measures ANOVA were applied to CMEP amplitude for low intensity and high intensity stimulation to examine main effects of drug ingestion and contraction intensity, or an interaction effect of drug condition by contraction intensity. To compare the effects of placebo and cyproheptadine ingestion on CMEP amplitudes from low and high intensity stimulation (i.e., placebo to cyproheptadine change score comparisons), separate two-way repeated measures ANOVA were applied to CMEP amplitude change scores for low intensity and high intensity stimulation to examine main effects of drug condition (placebo and cyproheptadine) and contraction

intensity, or an interaction effect of drug condition by contraction intensity. To compare differences in change scores for CMEP amplitudes induced by low and high intensity stimulation following placebo or cyproheptadine ingestion (i.e., change score comparisons between low intensity CMEPs and high intensity CMEPs for placebo and cyproheptadine), two-way repeated measures ANOVA was applied to CMEP amplitude change scores to examine main effects of stimulation intensity and contraction intensity, or an interaction effect of stimulation intensity by contraction intensity. Cohen's *d* was also calculated to determine the magnitude of effect between CMEP amplitude change scores for low and high intensity stimulation within the placebo and cyproheptadine conditions (negligible, $d < 0.2$; small, $d = 0.2$–0.5; moderate, $d = 0.5$–0.8). Tukey's multiple comparison test was applied to data if significant interaction effects were detected. CMEP data are presented in tables and figures as means ± 95% confidence intervals.

Similar to the analyses of evoked potentials, statistical tests were employed to assess any differences in motor unit measures between contraction intensities, testing sessions and drug conditions. These measures include motor unit discharge rates, recruitment threshold and derecruitment threshold. To account for the hierarchical nature of motor unit data, linear mixed effects models were developed for this analysis using the lmerTest package (Kuznetsova et al., 2017). Separate linear mixed effects models were used to assess any differences in baseline sessions between drug conditions. For this, models included fixed effects of testing day and contraction intensity, and an interaction effect of testing day by contraction intensity. Separate linear mixed effects models were also used to examine any differences in motor unit measures from baseline to post-pill ingestion across each contraction intensity. For these analyses, models included fixed effects of testing session (baseline and post-pill) and contraction intensity, and an interaction effect of testing session by contraction intensity. Relative change in discharge rates for placebo and cyproheptadine conditions were also assessed with fixed effects of drug condition and contraction intensity, and an interaction effect of drug condition by contraction intensity. All motor unit analyses included random intercepts for each participant, and units were nested within the same participants. For each linear mixed effects model, Satterthwaite's method was used to calculate degrees of freedom and *P*-values by comparing fitted models (i.e. with fixed effects) to a null model (i.e. without fixed effects). Tukey's multiple comparison tests were used if significant interaction effects were identified for any motor unit variables. Motor unit characteristics are presented in tables and figures as estimated marginal means (EMM) ± 95% confidence intervals calculated based on respective linear mixed effects models generated using the emmeans package (Lenth et al., 2021).

## Results

### Maximal voluntary contractions

Baseline peak torque was not significantly different between placebo and cyproheptadine testing conditions ($P = 0.142$). However, peak torque was reduced by ∼5% following cyproheptadine ingestion but not placebo ingestion (Fig. 3). There was a main effect of drug condition ($F_{(1, 11)} = 10.368$, $P = 0.008$), testing session ($F_{(1, 11)} = 12.599$, $P = 0.005$), and a significant interaction effect of drug condition by testing session ($F_{(1, 11)} = 10.068$, $P = 0.009$). *Post hoc* analysis revealed a significant difference in maximal torque between baseline and post-pill ingestion testing sessions in the cyproheptadine condition ($P = 0.003$), but no difference in the placebo condition ($P = 0.598$).

### RMS EMG during voluntary elbow flexions

RMS EMG during maximal contractions was not different between baseline sessions (placebo, $0.70 \pm 0.24$ mV; cyproheptadine, $0.70 \pm 0.25$ mV) across drug conditions ($t_{(1, 10)} = 1.58$, $P = 0.142$). Similarly, there were no

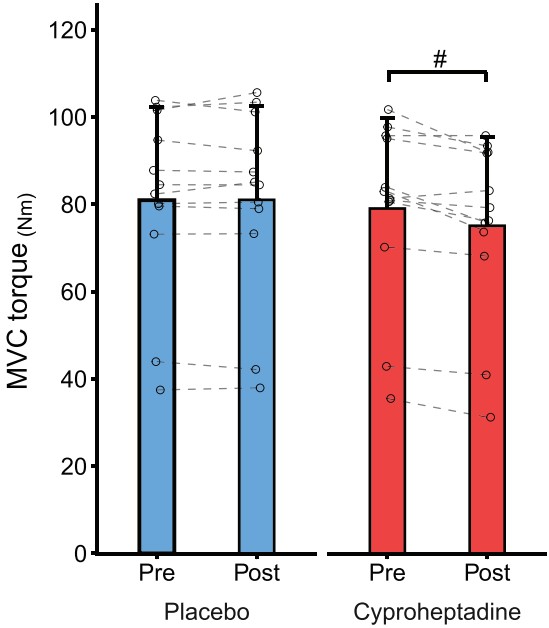

**Figure 3. Maximal elbow flexion torque during brief maximal voluntary contractions (MVC)**
MVC torque was measured pre- and post-pill ingestion for placebo (blue) and cyproheptadine (red) testing sessions. MVC torque is presented as group means ($n = 12$) for placebo and cyproheptadine, with error bars reflecting standard deviation. Individual data are presented as open circles for each testing session with dashed horizontal lines connecting individual participant MVC torque pre-pill (baseline) to post-pill for the respective testing condition (placebo or cyproheptadine). #Significant difference ($P = 0.003$) between pre- and post-pill testing sessions for the cyproheptadine condition. [Colour figure can be viewed at wileyonlinelibrary.com]

differences between baseline and post-pill ingestion (placebo, 0.71 ± 0.25 mV; cyproheptadine, 0.70 ± 0.25 mV) for placebo ($t_{(1, 10)}$ = −1.04, $P$ = 0.318) or cyproheptadine ($t$ = −0.32, $P$ = 0.758) conditions. However, during submaximal contractions, RMS EMG amplitude increased following cyproheptadine ingestion but not placebo (Table 1).

## Baseline motor unit characteristics during submaximal voluntary elbow flexions

From the 12 participants, one female participant was not included in motor unit statistical analyses as units could not be tracked from baseline to post-pill due to low yield from signal decomposition. Following decomposition, an average of 39 ± 16 and 32 ± 16 SD motor units were identified for each participant across contraction intensities for placebo and cyproheptadine conditions, respectively. Of these units identified, an average of 16 ± 6 and 13 ± 7 SD motor units were tracked for each participant from baseline to post-pill for placebo (176 total), and from baseline to post-pill for cyproheptadine (145 total) conditions, respectively. Baseline motor unit discharge rates did not differ between drug conditions ($P$ = 0.074), but did increase with contraction intensity [$F_{(2, 310)}$ = 60.250, $P$ < 0.001]. There was no interaction effect of testing day by contraction intensity ($P$ = 0.800) detected. Similarly, for baseline motor unit recruitment thresholds across placebo and cyproheptadine testing days, there was no main effect of testing day ($P$ = 0.123), a main effect of contraction intensity [$F_{(2, 313)}$ = 452.536, $P$ < 0.001], and no interaction effect of testing day by contraction intensity ($P$ = 0.889). For baseline derecruitment thresholds, there was no main effect of testing day ($P$ = 0.739), a main effect of contraction intensity [$F_{(2, 313)}$ = 255.688, $P$ < 0.001], and no interaction effect of testing day by contraction intensity ($P$ = 0.361). Collectively, these results suggest that motor units assessed on placebo and cyproheptadine testing sessions had similar firing characteristics prior to drug ingestion (Table 2), For representative data of the cyproheptadine condition, see Figure 4 below.

## Motor unit firing characteristics during submaximal voluntary elbow flexions following drug ingestion

During submaximal elbow flexions to 10, 20 and 30% of pre-pill (baseline) MVC, there were no differences of motor unit discharge rates, recruitment thresholds nor derecruitment thresholds following placebo ingestion (Table 2). However, following cyproheptadine ingestion, there was a significant increase in motor unit discharge rates (Fig. 5A), a reduction in recruitment threshold (Table 2) and an increase in derecruitment threshold

**Table 1. RMS EMG and $M_{MAX}$ amplitude during submaximal elbow flexions**

| | RMS EMG amplitude | | | | $M_{MAX}$ amplitude | | | |
| | Pre-pill (% of peak EMG) | | Post-pill change (post-pill − pre-pill, %) | | Pre-pill (mV) | | Post-pill change (post-pill − pre-pill, mV) | |
| | PLA | CYP | PLA | CYP | PLA | CYP | PLA | CYP |
|---|---|---|---|---|---|---|---|---|
| 10% MVC | 5.45 ± 2.71 | 5.10 ± 1.75 | −0.16 ± 1.01 | 0.79 ± 1.2 | 8.77 ± 2.53 | 8.85 ± 2.66 | 0.25 ± 0.44 | 0.41 ± 0.76 |
| 20% MVC | 12.35 ± 4.53 | 11.27 ± 3.23 | −0.02 ± 1.96 | 1.18 ± 2.78 | 8.99 ± 2.43 | 9.10 ± 2.72 | 0.22 ± 0.54 | 0.13 ± 0.83 |
| 30% MVC | 24.05 ± 9.40 | 21.62 ± 7.07 | −0.67 ± 3.15 | 4.19 ± 4.71 | 9.12 ± 2.70 | 9.26 ± 2.42 | 0.22 ± 0.68 | −0.18 ± 0.65 |
| Two-way repeated measures ANOVA | | | | | | | | |
| Drug condition | $F_{(1, 10)}$ = 2.79, $P$ = 0.126 | | $F_{(1, 10)}$ = 5.751, $P$ = 0.037* | | $F_{(1, 8)}$ = 0.411, $P$ = 0.539 | | $F_{(1, 8)}$ = 0.185, $P$ = 0.678 | |
| Contraction intensity | $F_{(2, 20)}$ = 55.50, $P$ < 0.001* | | $F_{(2, 20)}$ = 2.116, $P$ = 0.175 | | $F_{(2, 16)}$ = 2.845, $P$ = 0.121 | | $F_{(2, 16)}$ = 3.169, $P$ = 0.069 | |
| Interaction | $F_{(2, 20)}$ = 1.70, $P$ = 0.218 | | $F_{(2, 20)}$ = 11.4, $P$ < 0.001* | | $F_{(2, 16)}$ = 0.021, $P$ = 0.979 | | $F_{(2, 16)}$ = 1.398, $P$ = 0.276 | |

Two-way repeated measures ANOVA was used to identify differences in pre-pill RMS EMG amplitude and $M_{MAX}$ amplitude between placebo and cyproheptadine testing days and contraction intensities. Two-way repeated measures ANOVA was also used to identify any differences in change scores between placebo and cyproheptadine conditions for RMS EMG amplitude and $M_{MAX}$. Data are presented as means ± SD. Change scores were calculated by subtracting pre-pill from post-pill $M_{MAX}$ amplitude.
* Significant main effects identified from ANOVA tests.

**Table 2. Motor unit characteristics tracked from pre (baseline) to post ingestion for placebo and cyproheptadine**

| | Discharge rate (pps) | | Recruitment threshold (% of MVC) | | Derecruitment threshold (% of MVC) | |
|---|---|---|---|---|---|---|
| | Pre-pill | Post-pill | Pre-pill | Post-pill | Pre-pill | Post-pill |
| **Placebo** | | | | | | |
| Testing session | | | | | | |
| 10% MVC (39 units) | 12.06 [10.64, 13.50] | 12.19 [10.75, 13.62] | 8.10 [6.80, 9.42] | 7.29 [5.97, 8.62] | 3.01 [1.22, 4.80] | 2.98 [1.17, 4.79] |
| 20% MVC (78 units) | 14.14 [12.78, 15.49] | 14.13 [12.77, 15.48] | 16.07 [15.07, 17.08] | 16.36 [15.35, 17.37] | 10.31 [8.99, 11.62] | 9.57 [8.25, 10.88] |
| 30% MVC (58 units) | 15.93 [14.56, 17.31] | 15.64 [14.27, 17.01] | 23.97 [22.85, 25.09] | 23.01 [21.91, 24.12] | 19.64 [18.17, 21.12] | 18.76 [17.29, 20.22] |
| Linear mixed effects model | | | | | | |
| Session | $F_{(1, 175)} = 0.04, P = 0.836$ | | $F_{(1, 175)} = 2.34, P = 0.128$ | | $F_{(1, 175)} = 3.24, P = 0.134$ | |
| Contraction intensity | $F_{(2, 166)} = 36.25, P < 0.001^*$ | | $F_{(2, 170)} = 321.61, P < 0.001^*$ | | $F_{(2, 175)} = 137.62, P < 0.001^*$ | |
| Interaction | $F_{(2, 175)} = 0.80, P = 0.453$ | | $F_{(2, 175)} = 1.98, P = 0.141$ | | $F_{(2, 175)} = 0.85, P = 0.430$ | |
| **Cyproheptadine** | | | | | | |
| Testing session | | | | | | |
| 10% (36 units) | 10.95 [9.58, 12.31] | 12.59 [11.22, 13.95] | 8.76 [7.16, 10.36] | 8.61 [7.01, 10.21] | 2.60 [0.61, 4.58] | 4.03 [2.04, 6.02] |
| 20% (68 units) | 13.17 [11.92, 14.41] | 13.89 [12.64, 15.13] | 16.82 [15.47, 18.18] | 16.43 [15.08, 17.79] | 9.45 [7.87, 11.01] | 10.91 [9.34, 12.48] |
| 30% (41 units) | 15.38 [14.05, 16.72] | 16.49 [15.16, 17.82] | 25.71 [24.17, 27.24] | 23.85 [22.31, 25.39] | 21.58 [19.70, 23.46] | 22.81 [20.93, 24.69] |
| Linear mixed effects model | | | | | | |
| Session | $F_{(1, 145)} = 48.45, P < 0.001^*$ | | $F_{(1, 145)} = 8.46, P = 0.004^*$ | | $F_{(1, 145)} = 20.27, P < 0.001^*$ | |
| Contraction intensity | $F_{(2, 137)} = 31.17, P < 0.001^*$ | | $F_{(2, 138)} = 224.47, P < 0.001^*$ | | $F_{(2, 140)} = 150.73, P < 0.001^*$ | |
| Interaction | $F_{(2, 145)} = 2.70, P = 0.070$ | | $F_{(2, 145)} = 3.57, P = 0.031^*$ | | $F_{(2, 145)} = 0.06, P = 0.944$ | |

Linear mixed effects models were used to identify differences in motor unit discharge rates, recruitment threshold, and derecruitment threshold between pre-pill and post-pill testing sessions, and contraction intensities for the placebo and cyproheptadine conditions. *Significant main or interaction effects. Data are presented as estimated marginal means with lower and upper 95% CI. Pre-pill and post-pill motor unit discharge data for the cyproheptadine condition are presented in Fig. 5. pps, pulses per second.

(Table 2). For motor unit discharge rates in the cyproheptadine condition, there was a main effect of testing session [$F_{(1, 145)} = 48.450, P < 0.001$], a main effect of contraction intensity [$F_{(2, 137)} = 31.173, P < 0.001$], and no interaction effect of testing session by contraction intensity ($P = 0.070$). Relative change scores were calculated for motor unit discharge rates to assess any differences following pill ingestion across placebo and cyproheptadine testing days. There was a main effect of drug condition [$F_{(1, 319)} = 34.868, P < 0.001$], a main effect of contraction intensity [$F_{(2, 318)} = 3.364, P = 0.036$], and no interaction effect of drug by contraction intensity ($P = 0.161$) detected for discharge rate relative change scores (Fig. 5B).

## CMEP amplitude for low and high intensity stimulation

To establish whether low and high intensity cervicomedullary stimulation activates different proportions of the motoneurone pool, two-way repeated measures ANOVA were used to examine any differences in CMEP amplitude between low and high intensity stimulation across each contraction intensity in baseline conditions. There was a main effect of stimulation intensity [$F_{(1, 8)} = 155.258, P < 0.001$] and contraction intensity [$F_{(2, 16)} = 26.443, P < 0.001$]. There was no significant interaction effect of stimulation intensity by contraction intensity ($P = 0.668$) detected for baseline CMEP amplitude.

For baseline to post-pill comparisons, CMEPs produced by low intensity stimulation and high intensity stimulation were not affected by placebo ingestion. For placebo low intensity stimulation, there was no main effect of drug ingestion ($P = 0.068$). However, there was a main effect of contraction intensity [$F_{(2, 16)} = 19.264$, $P < 0.001$], but no interaction effect of drug ingestion by contraction intensity ($P = 0.434$). For placebo high intensity stimulation, there was no main effect of drug ingestion ($P = 0.123$). However, there was a main effect of contraction intensity [$F_{(2, 16)} = 18.300$, $P < 0.001$], but no interaction effect of drug ingestion by contraction intensity ($P = 0.593$). Cyproheptadine increased CMEP amplitude for both low and high intensity stimulation across each contraction intensity (Fig. 6). For cyproheptadine low intensity stimulation, there was a main effect of drug ingestion [$F_{(1, 8)} = 76.216$, $P < 0.001$] and contraction intensity [$F_{(2, 16)} = 17.901$, $P < 0.001$]. However, there was no significant interaction effect of drug ingestion by contraction intensity ($P = 0.645$). For high intensity stimulation in the cyproheptadine condition, there was a main effect of drug ingestion [$F_{(1, 8)} = 18.451$, $P = 0.003$] and contraction intensity [$F_{(2, 16)} = 23.466$, $P < 0.001$]. But there was no significant interaction effect of drug ingestion by contraction intensity ($P = 0.253$) detected for baseline CMEP amplitude.

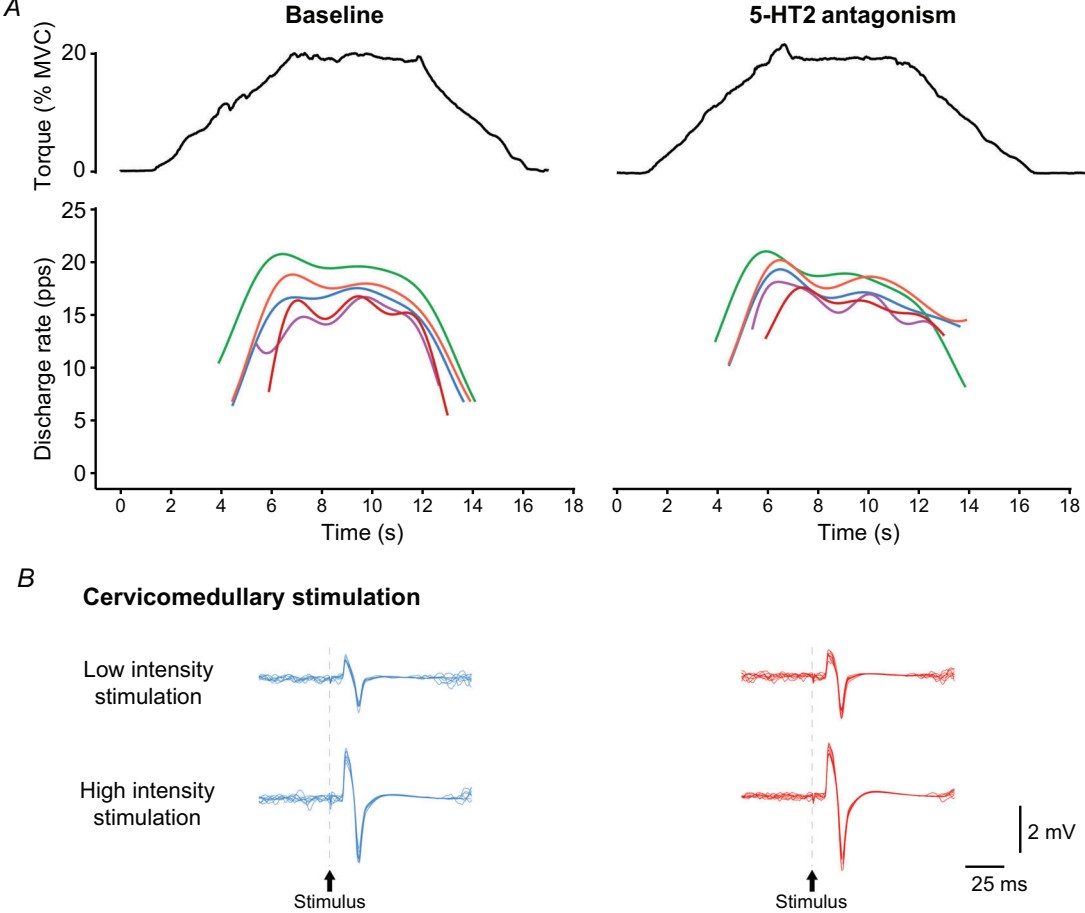

**Figure 4. Representative waveforms from one participant**
Participants performed a range of elbow flexion contraction tasks pre- (baseline) and post-pill ingestion of the 5-HT2 antagonist cyproheptadine. *A*, HDsEMG was used to track motor unit activity from baseline to post-pill. Support vector regressions (SVR) generated for the firing rates of motor units tracked from one participant during a 20% trapezoidal elbow flexion contraction are shown. Individual motor units are represented as the same colour for baseline and cyproheptadine sessions. *B*, cervicomedullary motor evoked potentials (CMEP) produced by low intensity and high intensity stimulation during different trials of the same contraction task were used to assess motoneurone excitability. Stimulations were delivered in the middle of the plateau phase of the contraction. Ten trials of low intensity and 10 trials of high intensity cervicomedullary stimulation were obtained from the same participant, and each trial of low and high intensity stimulation is overlaid for the respective stimulation intensity and testing condition (i.e. baseline and cyproheptadine) during contractions of 20% maximal torque. [Colour figure can be viewed at wileyonlinelibrary.com]

## Change in CMEP amplitude following placebo and cyproheptadine ingestion

When compared to placebo, cyproheptadine increased CMEP amplitude from both low and high intensity stimulation across each contraction intensity (Fig. 6C and D). For low intensity cervicomedullary stimulation change scores, there was a main effect of drug condition [$F_{(1, 8)} = 19.129, P = 0.002$] and a main effect of contraction intensity [$F_{(2, 16)} = 5.999, P = 0.011$] detected. However, there was no significant interaction effect of drug condition by contraction intensity ($P = 0.212$) detected for low intensity CMEP change scores. Thus, for low intensity stimulation, cyproheptadine change scores were larger than placebo, and this change was different across contraction intensities. For high intensity cervicomedullary stimulation change scores, there was a main effect of drug condition [$F_{(1, 8)} = 7.409, P = 0.026$], but no main effect of contraction intensity ($P = 0.100$), nor interaction effect of drug condition by contraction intensity ($P = 0.695$).

## Magnitude of change for CMEP amplitude produced by low and high intensity stimulation

Change scores from low and high intensity stimulation were also compared within each contraction intensity for placebo and cyproheptadine conditions. Cyproheptadine change scores were greater for low intensity cervicomedullary stimulation compared to

high intensity stimulation, and this difference between stimulation intensities reduced with an increase in contraction intensity (Fig. 7). For the placebo condition, there was no main effect of stimulation intensity ($P = 0.808$), nor contraction intensity ($P = 0.541$), and no interaction effect of stimulation intensity by contraction intensity ($P = 0.587$) detected for CMEP change scores. For cyproheptadine, there was a main effect of stimulation intensity [$F_{(1, 8)} = 6.354, P = 0.036$] and a main effect of contraction intensity [$F_{(2, 16)} = 3.813, P = 0.044$] detected for CMEP change scores. However, there was no interaction effect of stimulation intensity by contraction intensity ($P = 0.231$).

## Discussion

This project provides novel insight into the serotonergic modulation of human motoneurones during voluntary contractions. The 5-HT$_2$ receptor antagonist cyproheptadine caused a reduction in maximal elbow flexion torque. During contractions to 10%, 20% and 30% of pre-pill MVC, 5-HT$_2$ antagonism was found to: (1) increase EMG amplitude, (2) increase motor unit discharge rates across each contraction intensity, and (3) increase CMEP amplitudes from low and high intensity stimulation. These effects induced by 5-HT$_2$ antagonism were strongest during the lowest contraction intensity (10% of peak torque) and with low intensity cervicomedullary stimulation. Overall, these results suggest that additional excitatory drive is required to

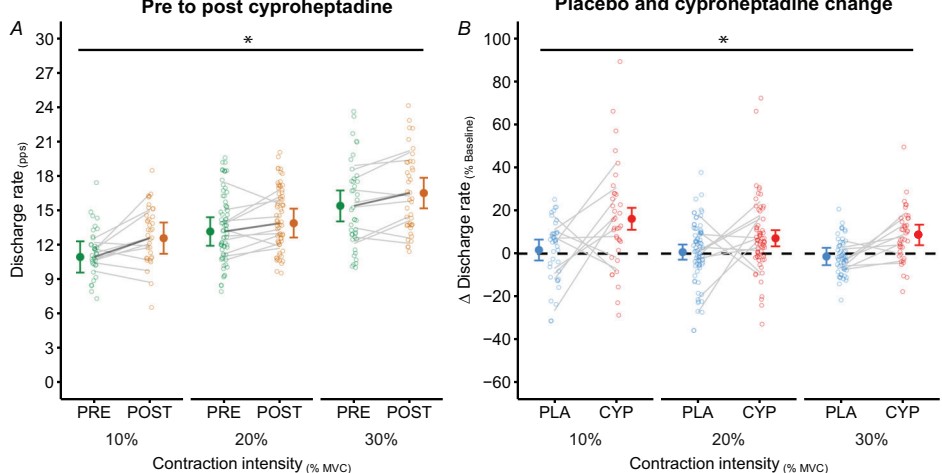

**Figure 5. Motor unit discharge rates**
Motor units were tracked from baseline to post-pill testing sessions for placebo and cyproheptadine conditions for 11 participants. *A*, discharge rates of 145 motor units in the cyproheptadine condition are presented in pulses per second (pps) for pre-pill (green circles) and post-pill (orange circles) testing sessions. *B*, comparisons of change in motor unit discharge rates (expressed as percentages of baseline) for the placebo (blue, 176 units) and cyproheptadine (red, 145 units) conditions. A positive change indicates an increase in discharge rate post-pill ingestion from baseline. Estimated marginal means and 95% CI are presented for group data, with individual motor units presented as open circles with light grey horizontal lines reflecting individual participant averages. *Significant main effect ($P < 0.05$) of testing session (*A*) and drug condition (*B*). [Colour figure can be viewed at wileyonlinelibrary.com]

achieve the same level of torque when 5-HT$_2$ receptors are antagonised.

### Excitatory drive to the motoneurone pool is increased to achieve the same level of submaximal torque with 5-HT$_2$ antagonism

It is well established that 5-HT$_2$ antagonism can reduce maximal torque produced by the elbow flexors (Henderson et al., 2023; Thorstensen et al., 2021, 2022). The current study provides further support for this finding, as there was a ∼5% reduction in peak torque following cyproheptadine ingestion. Despite this reduction in maximal torque post-pill ingestion, sub-maximal torque targets remained the same as pre-pill ingestion. During these 'torque matched' contractions, 5-HT$_2$ antagonism was found to increase RMS EMG amplitude of the biceps brachii. This finding could

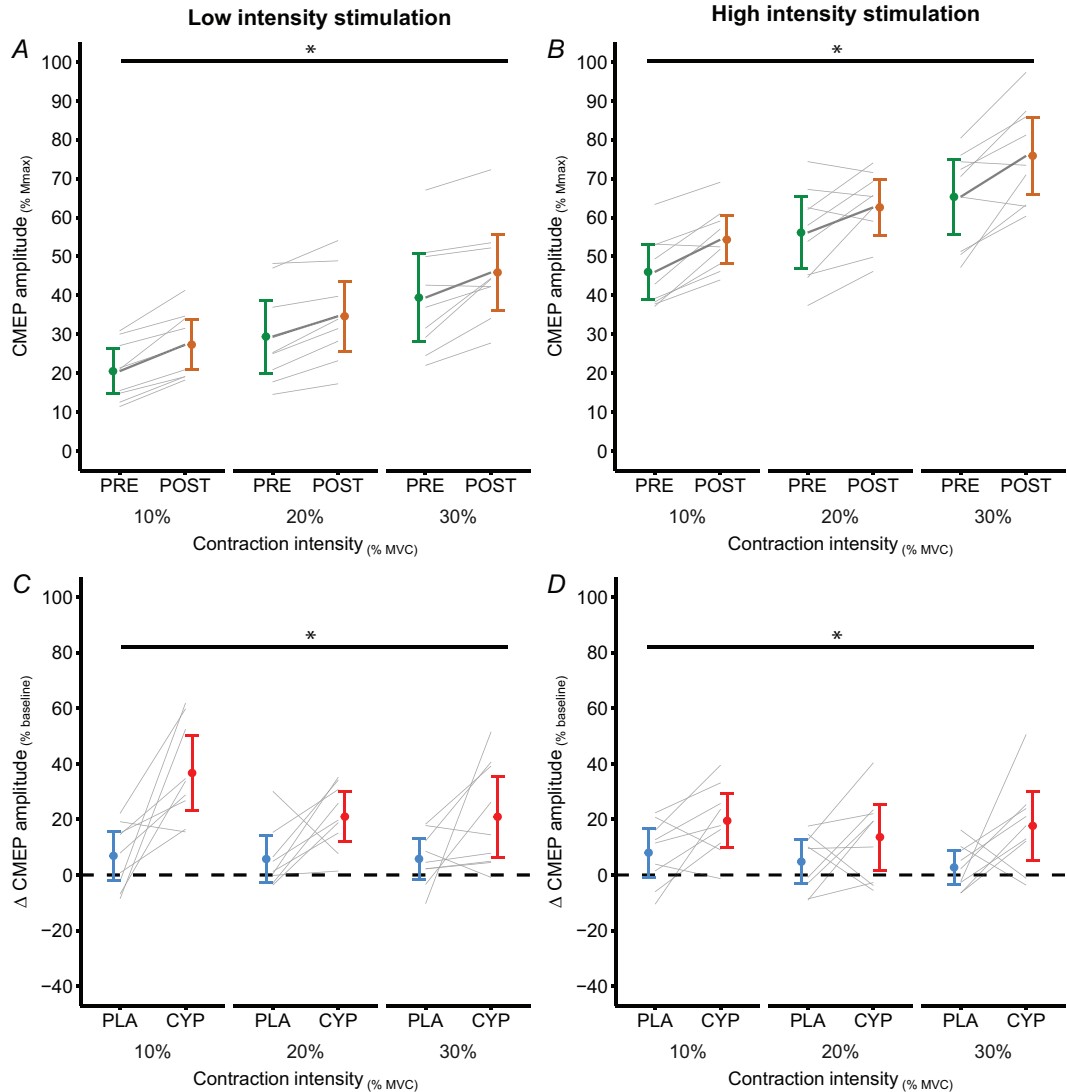

**Figure 6. CMEP amplitude for low and high intensity stimulation in the cyproheptadine condition**
*A*, for the cyproheptadine condition, pre-pill (green, PRE) and post-pill (orange, POST) CMEP amplitude is presented for low intensity cervicomedullary stimulation. *B*, CMEP amplitude is also presented for pre-pill and post-pill ingestion of cyproheptadine for high intensity cervicomedullary stimulation. Post-pill change scores were calculated for placebo (PLA, blue) and cyproheptadine (CYP, red) conditions for CMEPs produced by low and high intensity cervicomedullary stimulation. *C*, placebo and cyproheptadine change scores for CMEP responses from low intensity stimulation are presented. *D*, responses from high intensity stimulation are also presented. Change scores are presented as a percentage change from the respective baseline (pre-pill) measurement. All CMEP data are presented as means ± 95% CI with individual data presented as horizontal grey lines. *Significant main effect of drug ingestion (PRE *vs.* POST) for *A* and *B*, and a main effect of drug condition (PLA *vs.* CYP) for *C* and *D*. [Colour figure can be viewed at wileyonlinelibrary.com]

suggest that during the fixed-torque submaximal contractions, more motor units were recruited, or their firings were more synchronised, under conditions of 5-HT$_2$ antagonism. This increase in RMS EMG amplitude with 5-HT$_2$ antagonism could potentially be explained by the same mechanism causing an increase in motor unit firing rates also observed in this study.

We postulate that the observed increase in EMG amplitude and motor unit firing rates with 5-HT$_2$ antagonism can be, in part, attributed to an increase in excitatory drive to the motoneurone pool. With a reduction in maximal torque following cyproheptadine ingestion, the post-pill fixed submaximal torque targets were at a larger percentage of MVC for the drug condition. To match the submaximal contraction targets at a higher percentage of MVC, excitatory drive to elbow flexor motoneurones may be needed to increase motoneurone firing rates and/or to recruit additional motoneurones that are larger in size (Enoka & Duchateau, 2017; Heckman & Enoka, 2012; Henneman, 1957; Valencic et al., 2024). Thus, an increase in descending drive to the motoneurone pool may contribute to the increase

in RMS EMG amplitude and motor unit firing rates for the cyproheptadine condition. Interestingly, this increase in firing rates from motor units of the biceps brachii opposes previous findings of reduced firing rates of motor units from the tibialis anterior muscle with 5-HT$_2$ antagonism (Goodlich et al., 2023, 2024). Perhaps this previously identified reduction in discharge rates from lower limb motoneurones could be explained by the lack of differences in maximal dorsiflexion torque with 5-HT$_2$ antagonism. Given this previous work did not report any differences to maximal torque, and submaximal torque targets were set within each testing session, descending drive to the motoneurone pool probably remained similar across sessions.

The results of the current study indicate an increase in motor unit discharge rates even though the absolute level of submaximal torque did not change. Due to limitations of current HDsEMG technology, we are only able to assess a portion of motor units during contractions, and not the entire motoneurone pool. Thus, it is difficult to comment on the number of recruited motor units. However, recruitment thresholds were reduced for

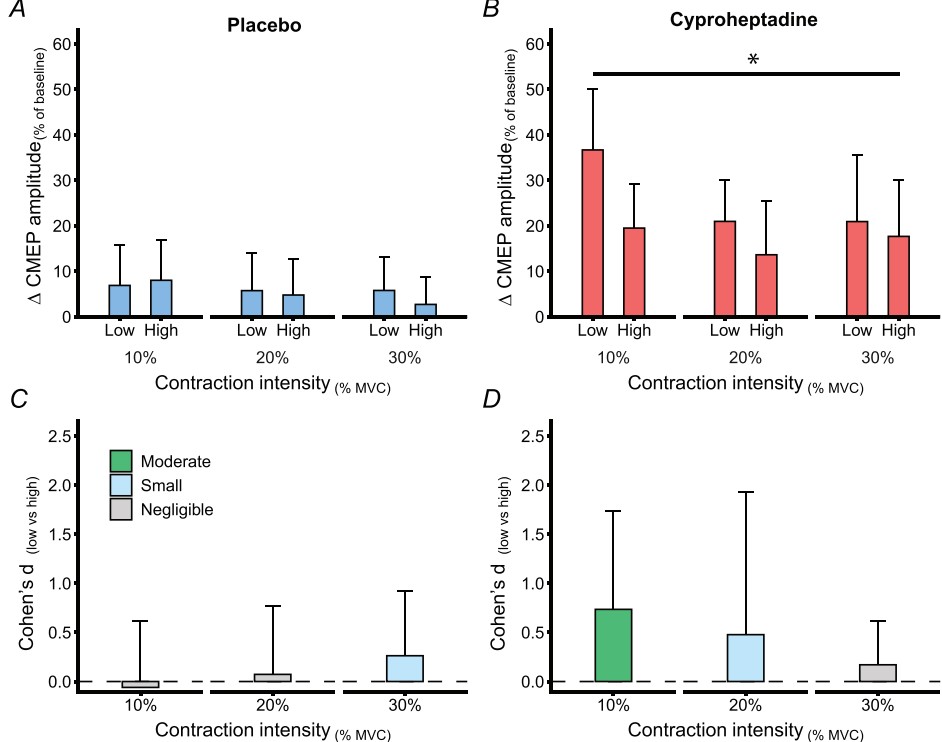

**Figure 7. Low intensity stimulation compared to high intensity stimulation change scores for CMEP amplitude**
*A* and *B*, change scores were calculated for placebo (*A*, blue) and cyproheptadine (*B*, red) conditions for CMEPs produced by low and high intensity cervicomedullary stimulation (*n* = 9, two female). Change scores are calculated as a percentage change from the respective baseline (pre-pill) measurement, and presented as group means ± 95% CI. *C* and *D*, the difference in change scores between low and high intensity stimulation in the placebo condition (*C*) and the cyproheptadine condition (*D*) presented as Cohen's *d*. A positive Cohen's *d* indicates that change in CMEP amplitude for low intensity stimulation is larger than high intensity stimulation. *Significant main effect (*P* < 0.05) of stimulation intensity. [Colour figure can be viewed at wileyonlinelibrary.com]

motor units under conditions of 5-HT$_2$ antagonism, with the largest reduction observed during 30% contractions. This suggests that higher threshold motoneurones were required earlier in the contraction to reach the same level of torque as pre-pill, and this effect was largest when greater torque production was required. This would suggest that either fewer low threshold motoneurones were recruited overall or less force was produced by the recruited units, and may explain the increase in discharge rates. Although 5-HT$_2$ receptors have been located on skeletal muscle in rats (Guillet-Deniau et al., 1997; Hajduch et al., 1999), the effects of these receptors on muscle contractility have not been reported. Thus, it is unknown whether the force produced by muscle fibres is affected by 5-HT$_2$ antagonism, and evidence for 5-HT effects on muscle fibre contractile properties is needed to provide support for the possible serotonergic effects at the muscle.

### Low and high intensity cervicomedullary stimulation during submaximal contractions

Electrical cervicomedullary stimulation recruits corticospinal axons at the pyramidal decussation to generate a single descending volley to motoneurones. Such stimulation activates motoneurones and produces a CMEP in upper limb muscles (McNeil et al., 2013; Petersen et al., 2002; Taylor, 2006; Taylor & Gandevia, 2004; Ugawa et al., 1991). During voluntary contractions, descending drive provides a level of facilitation to the motoneurone pool making some motoneurones fire and bringing other motoneurones closer to firing thresholds. This increase in descending drive with voluntary contractions positions more motoneurones in the subliminal fringe, and with the additional synaptic input provided by cervicomedullary stimulation, more motoneurones contribute to the evoked potential resulting in a larger CMEP amplitude. Consistent with previous findings, CMEP amplitude in the current study increased with an increase in contraction intensity (Henderson et al., 2023; McNeil et al., 2011; Pearcey et al., 2014; Yacyshyn et al., 2020). This increase in CMEP amplitude with contraction intensity was also consistent for smaller and larger proportions of the motoneurone pool (i.e. with low and high intensity stimulation).

Low intensity cervicomedullary stimulation is capable of activating large diameter corticospinal axons (Petersen et al., 2002), and the descending volley from cervicomedullary stimulation will likely recruit motoneurones in an orderly fashion from lower (smaller) to higher (larger) threshold depending on the stimulation intensity. As stimulation intensity increases, the CMEP latency decreases subtly, indicating that larger and faster conducting motoneurones are recruited (Taylor & Gandevia, 2004). Hence, with low intensity cervicomedullary stimulation, lower threshold motoneurones are primarily recruited. During baseline testing sessions in the current study, CMEPs produced by low intensity stimulation increased from ∼20% of $M_{MAX}$ amplitude during 10% contractions to ∼40% of $M_{MAX}$ amplitude during 30% contractions, and CMEPs produced by high intensity stimulation increased from ∼45% of $M_{MAX}$ during 10% contractions to ∼65% of $M_{MAX}$ during 30% MVC. Importantly, for all participants, within-session stimulation intensities for both the low and high intensity cervicomedullary stimulations were fixed and were not modified from pre-pill conditions. Considering the difference in CMEP amplitude between low and high intensity stimulation, we found that our higher intensity stimulations recruited more higher threshold motoneurones into the CMEP and will reflect the activity of both lower and higher threshold motoneurones.

### 5-HT$_2$ receptor antagonism increased CMEP amplitude produced by low and high intensity stimulation

Aligning with our previous work (Henderson et al., 2023), compared to the placebo condition, CMEPs evoked during voluntary contractions were larger under conditions of 5-HT$_2$ antagonism. The results from the current study provide novel insight to the effects of 5-HT$_2$ antagonism on smaller and larger proportions of the motoneurone pool, as this increase in CMEP amplitude was found for both smaller and larger CMEPs across all three contraction intensities. 5-HT$_2$ antagonism caused the largest change (∼35% increase from baseline) in CMEP amplitude during 10% MVC contractions with low intensity stimulation. During low intensity contractions, mostly low threshold motoneurones are voluntarily recruited to contribute to torque production. Likewise, mostly lower threshold motoneurones would contribute to the CMEP for low intensity stimulation during 10% contractions. Given that CMEP amplitudes are not affected by 5-HT$_2$ antagonism at rest (Henderson et al., 2023; Thorstensen et al., 2022), it is unlikely that direct effects on non-firing motoneurones contribute to the increased CMEP amplitude. Instead, it is possible that the increase in CMEP amplitude with cyproheptadine is caused by motoneurones close to firing thresholds (subliminal fringe), and that more motoneurones are readily available to contribute to the CMEP due to an increase in descending drive.

We previously speculated that a reduced PIC amplitude is a probable mechanism contributing to reductions in elbow flexion torque and an increase in CMEP amplitude (Henderson et al., 2023; Thorstensen et al., 2022). Given that 5-HT$_2$ receptors have strong effects on PICs (Harvey et al., 2006; Hounsgaard & Kiehn, 1989; Hounsgaard et al., 1988; Perrier & Delgado-Lezama, 2005; Perrier

& Hounsgaard, 2003), and previous human work has revealed reductions in estimates of PICs with $5\text{-HT}_2$ antagonism (Goodlich et al., 2023), it is possible that similar reductions in PICs occurred in the current study. Due to their voltage-dependence, PICs are only active when motoneurones receive sustained input to raise their membrane potentials near threshold. During 10% contractions, this applies to the small proportion of the motoneurone pool voluntarily recruited. With a decrease in PIC amplitude, extra synaptic current will be needed to raise the membrane voltage of the active motoneurones above the threshold for additional action potentials. That is, the active motoneurones will require more excitatory neural drive to maintain firing. In turn, extra drive to the motoneurone pool will cause a greater population of motoneurones to become closer to their firing thresholds, and with additional synaptic input from stimulation, these motoneurones will now discharge to contribute to the larger CMEP. Roughly similar increases in CMEP amplitude occur after cyproheptadine and with 10% MVC stronger voluntary contractions (see Fig. 6*A* and *B*). This suggests similar changes in the subliminal fringe in the two conditions. One interpretation is that the voluntary drive required to produce 10% MVC torque after cyproheptadine is close to that usually required for 20% MVC (i.e. almost double) while that require for 20% MVC after cyproheptadine is close to that usually required for 30% MVC (i.e. ∼50% increase). Previously, animal preparations involving cat motoneurones have revealed that lower threshold motoneurones rely more on intrinsic sources of depolarisation to maintain discharge rates. Hence, lower threshold motoneurones rely more heavily on dendritic PICs, as the amplification of firing induced by PICs causes these motoneurones to reach peak discharge rates more rapidly than higher threshold motoneurones, and smaller motoneurones sustain their firing for longer durations (Heckman et al., 2005). While it could be interpreted that these CMEP findings suggest lower threshold motoneurones may be more affected by $5\text{-HT}_2$ antagonism than higher threshold motoneurones, some consideration should be given to the application of relative change score metrics (percentage of control). While percentage change scores were employed to account for individual variability in baseline CMEP amplitude, small changes to raw amplitudes may reflect large percentage changes, particularly during low intensity contractions when CMEP amplitudes are small. Conversely, the alternative approach using absolute comparisons of change (post–pre) is highly dependent on baseline amplitudes which are larger during stronger contractions and does not account for changes in amplitude relative to contraction intensity. Therefore, interpretation of differential serotonergic effects on low and high threshold motoneurones remains speculative, as further exploration is needed for direct comparisons between low and high threshold motoneurones. Nevertheless, these CMEP findings provide insight into the effects of compensatory drive mediated by $5\text{-HT}_2$ antagonism across different levels of motoneurone activation, probed by low and high intensity stimulation.

It is important to consider that this possible mechanism of reduced PIC amplitude may not entirely explain the results of the current study and cannot be confirmed as it was not measured due to insufficient motor unit yield required for reliable estimates derived from $\Delta F$. If a reduction in PIC amplitude with $5\text{-HT}_2$ antagonism is the primary mechanism for the changes to outcome measures observed in this study, it seems plausible that larger effects would be observed during stronger contraction intensities when more motoneurones are active. Yet, drug effects were largest during the lowest intensity contractions. However, if lower threshold motoneurones are more strongly affected by $5\text{-HT}_2$ antagonism, the magnitude of effects would not proportionately increase with an increase in contraction intensity, as higher threshold motoneurones would potentially mask the more strongly affected lower threshold motoneurones. This is because firing rates of lower threshold motoneurones will be significantly increased, and a larger proportion of motoneurones are in the subliminal fringe and contributing to the CMEP during stronger contraction intensities. Additionally, an increase in motor unit firing rate would mean that the time these units are near threshold for firing would be reduced due to increased refractory period frequency, limiting the likelihood of these affected motoneurones contributing the CMEP (Martin et al., 2006; Matthews, 1996, 1999). Therefore, the serotonergic mechanisms underlying these results seem complex and difficult to explain. Nonetheless, the results of this study indicate that motoneurone excitability is affected by $5\text{-HT}_2$ modulation leading to compensatory drive to produce the same levels of torque.

### Considerations

While the present findings provide novel insights into the serotonergic modulation of the motoneurone pool in humans, some methodological considerations should be acknowledged. Firstly, cyproheptadine hydrochloride is a competitive antagonist of histamine $H_1$ receptors and $5\text{-HT}_2$ receptors, whilst also exhibiting minor anti-cholinergic properties by acting on muscarinic receptors. Antagonism of $H_1$ and $M_1$ receptors has previously been investigated with strong antihistamines, where promethazine hydrochloride was found to have no effect on motoneurone excitability following muscle contraction (Dempsey & Kavanagh, 2021), motor evoked potential amplitude during muscle contraction (Dempsey & Kavanagh, 2023), or short latency inhibition or facilitation with the muscle at rest (Di Lazzaro

et al., 2000). Although potential histaminic effects on motoneurone excitability from cyproheptadine ingestion are unlikely, the influence of these potential effects on the results of the current study cannot be completely ruled out. Secondly, examining the effects of serotonergic modulation across the motoneurone pool remains challenging due to methodological limitations. Although in theory, quantifying drug induced differences in motor unit discharge rates when categorised by recruitment thresholds could provide valuable insight to this research question, interpretations from these analyses need to be carefully considered as decomposition algorithms are more likely to detect larger motor units due to the low signal-to-noise ratio of low amplitude motor unit action potentials. Given that the current study had relatively low motor unit yield at each contraction intensity, and that the same motoneurones would exhibit different firing behaviour when receiving different levels of synaptic input, comparing the magnitude of change induced by 5-HT$_2$ antagonism in motor unit firing rates across recruitment thresholds was not feasible for this study. Therefore, comparisons of change for motor unit firing rates were restricted to 'within-contraction' analyses rather than analyses 'within recruitment thresholds'. Consequently, serotonergic effects on smaller proportions and larger proportions of the motoneurone pool are speculated based on CMEP responses to low and high intensity stimulation following cyproheptadine ingestion. While these conclusions on differential effects on smaller and larger proportions of the motoneurone pool remain speculative, this study provides novel insight into the serotonergic modulation of the motoneurone pool and presents valuable findings which can inform future research.

## Conclusion

This study is the first to assess the effects of 5-HT$_2$ receptor antagonism on motoneurone excitability using both HDsEMG and cervicomedullary stimulation in a single study, providing a unique insight into how 5-HT controls motoneurone activity in humans. This project found that motor unit discharge rates and CMEP amplitudes of the elbow flexors both increase with 5-HT$_2$ antagonism during submaximal elbow flexions when torque targets remain unchanged from baseline (pre-pill ingestion). It is likely that these changes are due to compensatory voluntary drive required to produce the same level of torque when 5-HT$_2$ receptors are antagonised.

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

## Additional information

### Data availability statement

The data supporting the results of this manuscript are available on request from the corresponding author.

### Competing interests

The authors declare that there are no competing interests.

### Author contributions

This experiment was performed at Griffith University, Gold Coast Campus, Queensland Australia. T.T.H., J.L.T., J.R.T. and J.J.K. conceptualised and designed the research. T.T.H. performed the experiment and analysed the data. T.T.H., J.L.T., J.R.T. and J.J.K. interpreted the results. T.T.H. drafted the manuscript, and T.T.H., J.L.T., J.R.T. and J.J.K. revised and edited the manuscript. All authors have read and approved the final version of this manuscript and agree to be accountable for all aspects of the work in ensuring that questions related to the accuracy or integrity of any part of the work are appropriately investigated and resolved. All persons designated as authors qualify for authorship, and all those who qualify for authorship are listed.

### Funding

No funding obtained to complete the work in this study.

### Acknowledgements

There authors would like to acknowledge all individuals who participated in this experiment and are thankful for their time.

Open access publishing facilitated by Griffith University, as part of the Wiley - Griffith University agreement via the Council of Australian University Librarians.

### Keywords

central nervous system, motoneurone, motor function, motor unit, neuromodulation, stimulation

## Supporting information

Additional supporting information can be found online in the Supporting Information section at the end of the HTML view of the article. Supporting information files available:

**Peer Review History**

