## [Peer Review History · The Journal of Physiology]

5-HT₂ antagonism indirectly increases motor unit discharge rate and cervicomedullary motor evoked potential amplitude during submaximal elbow flexions

Tyler T Henderson, Janet L Taylor, Jacob Thorstensen, and Justin J Kavanagh

DOI: 10.1113/JP288317

Corresponding author(s): Tyler Henderson (tyler.henderson@cdu.edu.au)

Review Timeline:

Submission Date:	08-Dec-2024
Editorial Decision:	26-Mar-2025
Revision Received:	22-May-2025
Editorial Decision:	11-Jun-2025
Revision Received:	12-Jun-2025
Accepted:	17-Jun-2025

Senior Editor: Richard Carson

Reviewing Editor: Jakob Škarabot

Transaction Report:

Dear Dr Henderson,

Re: JP-RP-2024-288317 "5-HT2 antagonism indirectly increases motor unit discharge rate and cervicomedullary motor evoked potential amplitude during submaximal elbow flexions" by Tyler T Henderson, Janet L Taylor, Jacob Thorstensen, and Justin J Kavanagh

Thank you for submitting your manuscript to The Journal of Physiology. It has been assessed by a Reviewing Editor and by 2 expert referees and we are pleased to tell you that it is potentially acceptable for publication following satisfactory major revision.

REVISION CHECKLIST:

We look forward to receiving your revised submission.

Yours sincerely,

Richard Carson
Senior Editor
The Journal of Physiology

REQUIRED ITEMS

- Your manuscript must include a complete Additional Information section, including competing interests; funding; author contributions and acknowledgements.

- Please upload separate high-quality figure files via the submission form.

EDITOR COMMENTS

Reviewing Editor:

Comments to the Author:

This study examined the effect of 5HT antagonism on biceps brachii motoneuron excitability with the aim of comprehensively assessing the effects on smaller and larger motoneurons through cervicomedullary stimulation and analysis of motor unit firing. Though both reviewers highlight several strengths, they both suggest revisions in the analytical approach. Specifically, the statistical analyses seem to obfuscate the potential effect (or lack thereof) using separate ANOVA models and % changes rather than the assessment of interactions. Furthermore, given the main research question, the analysis of motor unit firing should include the effects of recruitment threshold (either as a covariate or a categorical variable or both), rather than relying on torque amplitude, which is a poor proxy. Please also carefully consider your statements in your discussion regarding PICs and other motoneuron properties (e.g. synchronisation) - these statements are presently speculative, but they may be substantiated by additional analyses which are entirely possible with the available data (e.g. calculation of onset-offset hysteresis, coherence in time domain).

REFEREE COMMENTS

Referee #1:

The authors examine the effects of cyproheptadine (a 5-HT antagonist) on motor unit recruitment, discharge rates during isometric voluntary torque generation, and the amplitude of cervicomedullary motor evoked potentials (CMEPs). They report that cyproheptadine increases discharge rates, shifts recruitment thresholds, and increases CMEP amplitudes across various contraction intensities (10%, 20%, 30%) and stimulation intensities (10%, 35% MMax). While their findings significantly contribute to the literature, a few considerations and suggestions can be found below.

Major Comments/Considerations

There is considerable emphasis on the differential effects on lower and higher threshold motoneurons, even stating this as the purpose of the study (pg: 3, ln: 53). While this interpretation is appreciated, it is unclear that the methods employed adequately investigate this differential effect across recruitment. Perhaps the authors could temper their emphasis (as there are other novelties and additions to literature in this work) or carry out a more robust analysis if it is to be a major theme of the paper (see point 1 below).

In the CMEP experiments, insights into recruitment threshold effects are inferred by comparing changes across contraction and stimulation intensities. Specifically, lower-intensity contractions are expected to involve lower-threshold motoneurons, while lower-intensity stimulation is presumed to recruit motoneurons with relatively lower thresholds compared to higher-intensity stimulation, which would engage higher-threshold motoneurons. Although this reasoning seems sound, two potential issues are found with its application in the present manuscript:

1) This logic is not applied similarly to the interpretation of motor unit firing characteristics

Using the logic employed in the CMEP amplitude interpretations, wouldn't a drug by contraction intensity relation be expected if there were preferential effects on lower threshold MUs? Instead, there are no reported drug-induced discharge rate changes as a function of contraction intensity (pg. 19, ln 420 - 428). Furthermore, it seems as though Table 2 indicates that the largest change in recruitment thresholds occurs in the 30% contraction, not during the lower-intensity contractions.

Could the authors analyze motor unit (MU) discharge characteristics based on recruitment threshold? If sufficient data are available, MUs could be categorized by relative recruitment thresholds (e.g., 0-10%, 10-20%, 20-30%) and included as factors in their mixed model. This approach could enable the estimation of recruitment threshold effects and allow for post hoc testing to assess whether the observed changes are more pronounced in lower-threshold MUs.

2) Implementation of %change metrics could obfuscate interpretation

While it is appreciated that % change (post-pre/pre) is often a helpful metric, it is difficult to contextualize the use of this metric with the authors' "subliminal fringe" interpretation. When analyzing raw amplitudes, there is no significant interaction between drug and contraction intensity for either high or low intensity stimulation (pg. 22, ln. 455 - 461). Does this not imply that the magnitude of units in the subliminal fringe is uniformly increased by cyproheptadine irrespective of contraction intensity, or are you attempting to normalize by motoneuron size? A greater emphasis on why the change metrics are critical for the subliminal fringe interpretation is needed, as it does not appear readily apparent.

Minor Comments/Considerations

Have the authors given any consideration to the potential histaminergic effects of cyproheptadine? Histamine does modulate motor behaviors across various animal models. Could histaminergic-mediated modulation play a role in any of the reported findings?

The word "indirectly" is used in the title, but only in the title. It is unclear what the authors intended to portray with this. While it is presumed that this indirect nature refers to the changes in necessary excitatory drive induced by PIC changes, the authors should better highlight what they mean by indirect effects in the text (introduction and/or discussion).

What was the rationale used for the dosage (8 mg) of cyproheptadine, as well as a static dose across all participants?

Were smooth or instantaneous estimates of discharge used for the quantification of mean discharge rate? The authors should clarify this (somewhere around pg. 11, ln. 292), since they show smoothed discharge rates in Figure 3A. Also, if

smoothed discharge rates were used, were the hyperparameters suggested by Beauchamp et. al., used or something different?

How many units on average per participant were decomposed?

Figure 3: Could you indicate matched units from pre- to post-pill ingestion with matching colors here? An increase in discharge rate is reported consistently in the results, but the example shown makes that hard to appreciate.

Table 2 & Figure 4: It would be helpful if the EMM and LMM results from Figure 4 were shown quantitatively in Table 2. I know this is a personal preference, but it makes interpretation far easier if you can appreciate both the qualitative and quantitative aspects.

Pg.24, ln. 485: "($p=0.0.587$)"

Pg. 26, ln 514: If motor unit synchronization is thought responsible, couldn't you investigate this with coherence or metrics of shared synaptic input (Negro et. al.,).

Pg. 29, ln. 600: Double period after amplitude

Pg. 29, ln. 611: PICs being voltage-dependent does not necessitate voluntary drive for their activation. Input that drives the membrane suprathreshold long enough to allow activation of the PIC would likely be sufficient, irrespective of its voluntary or involuntary nature.

Referee #2:

- This paper builds on recent findings from the same research group, which demonstrated that 5-HT2 antagonism increases cervicomedullary motor-evoked potentials (CMEPs) in the right biceps brachii. Expanding on these results, the present study investigates the effects of 5-HT2 antagonism on biceps brachii motor neuron excitability by combining high-density surface EMG (HDsEMG) recordings with cervicomedullary stimulation during voluntary contractions. Overall, the study addresses an interesting and relevant research question and is well conducted, with a clearly identified research gap, well-defined purpose, and explicit hypotheses. However, several methodological and analytical concerns must be addressed to strengthen the conclusions. I outline these points below.

- One of the most critical concerns is the recording of CMEPs. What was the approximate inter-electrode distance of the bipolar EMG electrodes (Lines 178-179)? It appears that the inter-electrode distance may have been relatively large. While a greater inter-electrode distance increases conduction volume and captures a more representative response of muscle excitation, it also raises the risk of crosstalk from adjacent muscles (e.g., brachialis), particularly during voluntary contractions. Did the authors assess potential crosstalk from other flexor muscles? This is crucial because while motor unit activity was recorded exclusively from the biceps brachii, there is a risk that CMEPs were recorded from both the biceps brachii and other muscles, which could compromise data interpretation. A straightforward way to address this concern would be to compare responses obtained from the bipolar EMG system with a bipolar configuration extracted from the electrode grid. If these responses are similar, it would confirm that CMEPs were exclusively recorded from the biceps brachii.

- Additional details regarding brachial plexus stimulation are needed. What were the dimensions of the cathode and anode? How was the stimulation intensity for Mmax determination selected (e.g., staircase protocol)? A key concern is that the stimulation intensity remained unchanged across sessions. Surface electrode stimulation of the brachial plexus is highly sensitive to slight variations in electrode positioning and skin-electrode contact. Therefore, it is essential to report intra-session reliability measures (intraclass correlation coefficients) to ensure that M-wave amplitudes elicited at the same intensity were reproducible across sessions. Similarly, intra-session reliability should be reported for CMEP responses, as the same stimulation intensities ("low" and "high" stimulation) were used across sessions.

- It is unclear why separate ANOVAs were used to compare the effects of placebo vs. cyproheptadine conditions and the effects of low vs. high-intensity stimulation on CMEPs. Why was stimulation intensity not included as a cofactor in the first analysis? A linear mixed-effects model, as used in the motor unit analysis, could also be applied to assess evoked responses, potentially providing a more comprehensive statistical approach.

- The rationale behind selecting 15%, 25%, and 35% MVC for calibration contractions (motor unit filter extraction) while using different torque levels (10%, 20%, and 30% MVC) for ramped contractions is unclear. Please provide an explanation for this discrepancy.

- While a sample size of approximately ten participants is common in this field, it remains relatively small. Please provide a sample size calculation to justify the adequacy of the chosen sample. Additionally, there is no mention of whether written informed consent was obtained from participants. Please ensure this information is included.

- To improve clarity, please include schematic figures illustrating electrode placement for both stimulation and recording. This would help readers better understand electrode positioning, particularly for cervicomedullary stimulation.

Minor comments:

Line 93: Please introduce the term "CMEP" in full before using the abbreviation.

Lines 140-141: Please include a reference to support this statement.

Lines 152-153: Provide additional details on participant characteristics, including height and weight.

Lines 178-179: How were the midpoint of the muscle belly and the distal tendon of the biceps identified? Was ultrasound or palpation used? Please clarify.

Lines 187-189: Please specify the gain settings for the HDsEMG signals.

Lines 260-262: The description of "initial deflection in the bipolar EMG signal to the second crossing of 0 mV" is unclear. Please provide further details for clarity.

Lines 398-399: Include the average number of tracked motor units per participant.

END OF COMMENTS

The authors examine the effects of cyproheptadine (a 5-HT antagonist) on motor unit recruitment, discharge rates during isometric voluntary torque generation, and the amplitude of cervicomedullary motor evoked potentials (CMEPs). They report that cyproheptadine increases discharge rates, shifts recruitment thresholds, and increases CMEP amplitudes across various contraction intensities (10%, 20%, 30%) and stimulation intensities (10%, 35% MMax). While their findings significantly contribute to the literature, a few considerations and suggestions can be found below.

Major Comments/Considerations

There is considerable emphasis on the differential effects on lower and higher threshold motoneurons, even stating this as the purpose of the study (pg: 3, ln: 53). While this interpretation is appreciated, it is unclear that the methods employed adequately investigate this differential effect across recruitment. Perhaps the authors could temper their emphasis (as there are other novelties and additions to literature in this work) or carry out a more robust analysis if it is to be a major theme of the paper (see point 1 below).

In the CMEP experiments, insights into recruitment threshold effects are inferred by comparing changes across contraction and stimulation intensities. Specifically, lower-intensity contractions are expected to involve lower-threshold motoneurons, while lower-intensity stimulation is presumed to recruit motoneurons with relatively lower thresholds compared to higher-intensity stimulation, which would engage higher-threshold motoneurons. Although this reasoning seems sound, two potential issues are found with its application in the present manuscript:

1) This logic is not applied similarly to the interpretation of motor unit firing characteristics

Using the logic employed in the CMEP amplitude interpretations, wouldn't a drug by contraction intensity relation be expected if there were preferential effects on lower threshold MUs? Instead, there are no reported drug-induced discharge rate changes as a function of contraction intensity (pg. 19, ln 420 – 428). Furthermore, it seems as though Table 2 indicates that the largest change in recruitment thresholds occurs in the 30% contraction, not during the lower-intensity contractions.

Could the authors analyze motor unit (MU) discharge characteristics based on recruitment threshold? If sufficient data are available, MUs could be categorized by relative recruitment thresholds (e.g., 0–10%, 10–20%, 20–30%) and included as factors in their mixed model. This approach could enable the estimation of recruitment threshold effects and allow for post hoc testing to assess whether the observed changes are more pronounced in lower-threshold MUs.

2) Implementation of %change metrics could obfuscate interpretation

While it is appreciated that % change (post-pre/pre) is often a helpful metric, it is difficult to contextualize the use of this metric with the authors' "subliminal fringe" interpretation. When analyzing raw amplitudes, there is no significant interaction between drug and contraction intensity for either high or low intensity stimulation (pg. 22, ln. 455 – 461). Does this not imply that the magnitude of units in the subliminal fringe is uniformly increased by cyproheptadine irrespective of contraction intensity, or are you attempting to normalize by motoneuron size? A greater emphasis on why the change metrics are critical for the subliminal fringe interpretation is needed, as it does not appear readily apparent.

Minor Comments/Considerations

Have the authors given any consideration to the potential histaminergic effects of cyproheptadine? Histamine does modulate motor behaviors across various animal models. Could histaminergic-mediated modulation play a role in any of the reported findings?

The word “indirectly” is used in the title, but only in the title. It is unclear what the authors intended to portray with this. While it is presumed that this indirect nature refers to the changes in necessary excitatory drive induced by PIC changes, the authors should better highlight what they mean by indirect effects in the text (introduction and/or discussion).

What was the rationale used for the dosage (8 mg) of cyproheptadine, as well as a static dose across all participants?

Were smooth or instantaneous estimates of discharge used for the quantification of mean discharge rate? The authors should clarify this (somewhere around pg. 11, ln. 292), since they show smoothed discharge rates in Figure 3A. Also, if smoothed discharge rates were used, were the hyperparameters suggested by Beauchamp et. al., used or something different?

How many units on average per participant were decomposed?

Figure 3: Could you indicate matched units from pre- to post-pill ingestion with matching colors here? An increase in discharge rate is reported consistently in the results, but the example shown makes that hard to appreciate.

Table 2 & Figure 4: It would be helpful if the EMM and LMM results from Figure 4 were shown quantitatively in Table 2. I know this is a personal preference, but it makes interpretation far easier if you can appreciate both the qualitative and quantitative aspects.

Pg.24, ln. 485: “(p=0.0587)”

Pg. 26, ln 514: If motor unit synchronization is thought responsible, couldn't you investigate this with coherence or metrics of shared synaptic input (Negro et. al.,).

Pg. 29, ln. 600: Double period after amplitude

Pg. 29, ln. 611: PICs being voltage-dependent does not necessitate voluntary drive for their activation. Input that drives the membrane suprathreshold long enough to allow activation of the PIC would likely be sufficient, irrespective of its voluntary or involuntary nature.

Response to referees:

EDITOR COMMENTS

Reviewing Editor:

This study examined the effect of 5HT antagonism on biceps brachii motoneuron excitability with the aim of comprehensively assessing the effects on smaller and larger motoneurons through cervicomedullary stimulation and analysis of motor unit firing. Though both reviewers highlight several strengths, they both suggest revisions in the analytical approach. Specifically, the statistical analyses seem to obfuscate the potential effect (or lack thereof) using separate ANOVA models and % changes rather than the assessment of interactions. Furthermore, given the main research question, the analysis of motor unit firing should include the effects of recruitment threshold (either as a covariate or a categorical variable or both), rather than relying on torque amplitude, which is a poor proxy. Please also carefully consider your statements in your discussion regarding PICs and other motoneuron properties (e.g. synchronisation) - these statements are presently speculative, but they may be substantiated by additional analyses which are entirely possible with the available data (e.g. calculation of onset-offset hysteresis, coherence in time domain).

After carefully reviewing the comments of each Referee, we were able to address all concerns by providing a clarification in the response letter and making direct changes to the manuscript. We have also performed the suggested statistical approach for CMEP responses and motor unit data. Although, these analyses have reinforced several interpretations from our original discussion and conclusion, the analyses have not been added to the revised manuscript for several reasons described in the response to the referee comments below. Instead, as suggested by Referee #1, we have tempered the emphasis on differentiating serotonergic effects between lower and higher threshold motoneurons. We also acknowledge that statements of motor unit synchronisation and PICs remain speculative. Due to the relatively low motor unit yield for units across contraction intensities, onset-offset hysteresis and coherence measures may not be reliable measures that reflect the true nature of the motor unit pool, and these analyses have not been performed. This has now been addressed in the discussion, and associated statements have been reframed to better reflect our approach and analyses.

REFEREE COMMENTS

Referee #1:

The authors examine the effects of cyproheptadine (a 5-HT antagonist) on motor unit recruitment, discharge rates during isometric voluntary torque generation, and the amplitude of cervicomedullary motor evoked potentials (CMEPs). They report that cyproheptadine increases discharge rates, shifts recruitment thresholds, and increases CMEP amplitudes across various contraction intensities (10%, 20%, 30%) and stimulation intensities (10%, 35% MMax). While their findings significantly contribute to the literature, a few considerations and suggestions can be found below.

We thank Referee #1 for their insightful comments.

Major Comments/Considerations

There is considerable emphasis on the differential effects on lower and higher threshold motoneurons, even stating this as the purpose of the study (pg: 3, ln: 53). While this interpretation is appreciated, It is unclear that the methods employed adequately investigate this differential effect across recruitment. Perhaps the authors could temper their emphasis (as there are other novelties and additions to literature in this work) or carry out a more robust analysis if it is to be a major theme of the paper (see point 1 below).

To address this concern, phrases have now been altered to reduce the emphasis on lower vs higher threshold comparisons, and a considerations section has been added to the discussion. While we acknowledge that the interpretations of differential effects across recruitment threshold should remain speculative, we believe that the findings of this study provide novel insights that will inform future investigations in this area. Therefore, this section of the discussion has remained, but the text has been altered and considerations have been added to reflect the explorative nature of these interpretations.

To more appropriately align with the project aims and results, alterations have been made to the purpose statement, introduction and discussion. Notable amendments are:

- Key points (lines 34 – 36): the key point regarding lower and higher threshold motoneurons has been replaced with “These findings suggest that compensatory voluntary drive is required to achieve the same torque level with 5-HT₂ receptor antagonism, indirectly enhancing motoneurone activity and excitability of the motoneurone pool.”
- Abstract (line 40): “Therefore, the purpose of this study was to assess the effects of 5-HT₂ antagonism on motor unit discharge characteristics of the biceps brachii and evoked responses to cervicomedullary stimulation.”
- Introduction (line 103): the ‘differential effects’ on smaller and larger proportions of the motoneurone pool are introduced as a secondary purpose: “A secondary purpose of this study was to assess the effects of 5-HT₂ antagonism on different proportions of the motoneurone pool.”

- The Discussion section now includes direct considerations for these interpretations (see comments below).
- Conclusion (line 692): “This project found that motor unit discharge rates and CMEP amplitudes of the biceps brachii both increase with 5-HT₂ antagonism during submaximal elbow flexions when torque targets remain unchanged from baseline (pre-pill ingestion). It is likely that these changes are due to compensatory voluntary drive required produce the same level of torque when 5-HT₂ receptors are antagonised.”

In the CMEP experiments, insights into recruitment threshold effects are inferred by comparing changes across contraction and stimulation intensities. Specifically, lower-intensity contractions are expected to involve lower-threshold motoneurons, while lower-intensity stimulation is presumed to recruit motoneurons with relatively lower thresholds compared to higher-intensity stimulation, which would engage higher-threshold motoneurons. Although this reasoning seems sound, two potential issues are found with its application in the present manuscript:

1) This logic is not applied similarly to the interpretation of motor unit firing characteristics

Using the logic employed in the CMEP amplitude interpretations, wouldn't a drug by contraction intensity relation be expected if there were preferential effects on lower threshold MUs? Instead, there are no reported drug-induced discharge rate changes as a function of contraction intensity (pg. 19, ln 420 - 428).

This is something we have strongly considered when designing the experiments and when interpreting the findings. Thus, changes to CMEP amplitudes are used to complement the motor unit data and provide further insight to a wider range of the motoneurone pool. This is because CMEP amplitude reflects the excitability of the voluntarily recruited units as well as those close to firing thresholds. With low intensity stimulation during low intensity contractions, this will be mostly lower threshold units, whereas the higher stimulation intensity during stronger contractions would reflect a larger population of both lower and higher threshold units. Given that CMEP responses likely reflects a broader variety of lower threshold units compared to the motor unit recordings, additional analyses are performed on the CMEP data. Together, motor unit recordings and CMEP amplitudes are used as complementary methods to provide further insights to motoneurone excitability across different contraction tasks, and CMEP results are used to speculate on the effects across different proportions of the motoneurone pool.

While it could be expected that a drug by contraction intensity interaction should be present in the current findings for motor unit firing characteristics, there are some considerations that need to be addressed when inferring these effects on changes to motor unit discharge rates and recruitment thresholds. Notably, a large portion of ‘low-threshold’ motor units are recruited very early in the contraction (< 5% MVC), and due to low signal to noise ratios with small action potentials, the HDsEMG decomposition is not sensitive to detect these low

threshold motor units. In the current study, there were only four units identified below 10% MVC with the mean recruitment thresholds for each contraction intensity close to the upper limit of the target torque (RT ~9%, ~17% and ~25% MVC for torque targets of 10%, 20% and 30% MVC), and these identified units likely mask the effects of cyproheptadine on lower threshold units that were not identified from the decomposition. Therefore, inferring differences to lower and higher threshold motor units based on the motor unit recordings from HDsEMG has been approached with caution in the current study. Instead, motor unit recordings during different contraction intensities were used to examine motor unit firing characteristics during each contraction task (i.e., firing behaviour during different levels of excitatory drive), and not to infer changes across recruitment thresholds. Nonetheless, changes have been made to the text of the manuscript to reduce the emphasis of these potential differences between lower and higher threshold units.

Furthermore, it seems as though Table 2 indicates that the largest change in recruitment thresholds occurs in the 30% contraction, not during the lower-intensity contractions.

We appreciate the Referee's observation of the changes to recruitment thresholds presented in Table 2. Indeed, the largest change in recruitment threshold is observed during a 30% contraction (~ -1.85%), and this has now been included in our interpretations in the discussion (lines 535 - 539). However, we have approached this result with caution for similar reasons mentioned in the comment above regarding motor unit decomposition. Additionally, during higher intensity contractions, recruitment continues throughout the ramp (i.e., wider recruitment window), and small shifts in onset firing can result in larger changes in recruitment threshold. During the 10% contractions, these shifts in onset firing are restricted to 10% and may not reflect the recruitment thresholds in a similar fashion to higher intensity contractions. This would cause a larger observed difference in recruitment thresholds which is likely to be primarily due to the contraction task and not the magnitude of effects of 5-HT modulation across the motoneurone pool. Moreover, interpretations on differences between lower and higher threshold units can similarly be applied to derecruitment thresholds, whereby the largest differences due to cyproheptadine ingestion are during 10% and 20% MVCs. Therefore, these results are included in the interpretations in the discussion section but do not form a key theme of the paper.

Could the authors analyze motor unit (MU) discharge characteristics based on recruitment threshold? If sufficient data are available, MUs could be categorized by relative recruitment thresholds (e.g., 0-10%, 10-20%, 20-30%) and included as factors in their mixed model. This approach could enable the estimation of recruitment threshold effects and allow for post hoc testing to assess whether the observed changes are more pronounced in lower-threshold MUs.

We thank the Referee for their comment. As suggested, we have completed additional analyses to investigate the effects of cyproheptadine on motor units when categorised by their relative recruitment thresholds. However, there are significant challenges with the interpretations of these results, and we feel that these analyses do not appropriately examine motor unit firing characteristics when categorised by recruitment thresholds. Therefore, we

have included these suggested analyses in this Response Letter in the interest of the Referee but have not included these analyses in the revised manuscript.

To perform these analyses, motor units were categorised by their respective recruitment thresholds, and units were pooled into bin ranges of 0-10%, 11-20%, and 21-30%. First, a LMM was performed on this data with fixed effects of drug condition (placebo and cyproheptadine), testing session (pre-pill and post-pill), motor unit recruitment threshold (0-10%, 11-20%, and 21-30%), and their interactions on motor unit discharge rates. There were random intercepts for each participant, with units nested within the same participants. A subsequent model was also performed to include the fixed effect of contraction intensity. Similar models were also performed for change score comparisons of motor unit discharge rates, with separate models for recruitment thresholds treated as a covariate (continuous) and categorical (bins). Detailed results of each model are provided in the 'Statistical Analysis (for reviewers)' spreadsheet attached to this submission. To summarise, the results of these models closely align with models used previously. Whereby:

- Cyproheptadine increased absolute discharge rates compared to pre-pill ($p < 0.001$) with MU's categorised by bins
- Cyproheptadine increased discharge rate change scores compared to placebo ($p < 0.001$) with MU's categorised by bins
- Discharge rate increased with recruitment threshold for change score comparisons ($p < 0.001$), with MU's categorised as continuous
- However, there was no interaction effects involving recruitment thresholds ($p > 0.05$)

In the interest of the Referee's request, explorative post-hoc analyses were also completed. For change scores when categorised by recruitment thresholds, explorative post-hoc analyses revealed differences between placebo and cyproheptadine at each recruitment threshold bin (0-10%, 11-20%, 21-30%), with the largest difference observed in the lowest recruitment thresholds (0-10% MVC, $p = 0.001$). Given that there was no three-way interaction (i.e., drug condition by recruitment threshold by contraction intensity), these additional post-hoc analyses are not used to directly infer larger differences in lower threshold motoneurons.

We appreciate the suggestion to include these analyses in the manuscript; however, there are several challenges with this inclusion. Notably, the decomposition of HDsEMG signals predominantly identifies larger threshold units within each contraction (i.e., units recruited closer to the upper limit of the force target), which has occurred in the current dataset as the average recruitment threshold for units identified during 10%, 20% and 30% contractions are ~9% MVC, ~17% MVC, ~25% MVC, respectively. Another consideration is that motor units are likely to have different firing patterns during different levels of force production due to changes in levels of synaptic input, and motor units in this study were not tracked across contraction intensities. Therefore, units would still need to be grouped within each contraction intensity, but within each contraction intensity, motor units are mostly clustered around higher recruitment thresholds. Additionally, these models become statistically complex as they apply three or four factors (similar to a three- and four-way repeated measures ANOVA), and three-way interaction effects are needed to determine changes across recruitment thresholds (requiring very high statistical power only achievable with a very large sample size, and typically not appreciated by the neurophysiology community). To address these challenges, separate linear mixed models for each contraction intensity were also completed (see Figure 1 below). However, due to the bias towards identifying higher threshold units compared to lower threshold units within each contraction task, the variability

for the estimated change in discharge rate across recruitment threshold is considerably larger at the lower recruitment thresholds. Ultimately, this has significantly reduced the accuracy and reliability of the individual models. While valuable in theory, this suggested statistical approach is not suitable to reflect differential effects of serotonin across the motoneurone pool in the current study and we have not included these analyses in the manuscript. However, in acknowledgment of this suggestion, we have added a ‘*Considerations*’ section to the manuscript (line 658).

Figure 1: To account for different contraction tasks, motor units were grouped by recruitment threshold within each contraction intensity (10%, 20% and 30% MVC). A linear mixed model was applied to each contraction intensity to provide an estimate of change in discharge rate for placebo (blue) and cyproheptadine (red) conditions. The estimates are plotted as lines with 95% CI and individual change scores for each unit are presented as circles.

2) Implementation of %change metrics could obfuscate interpretation

While it is appreciated that % change (post-pre/pre) is often a helpful metric, it is difficult to contextualize the use of this metric with the authors' "subliminal fringe" interpretation. When analyzing raw amplitudes, there is no significant interaction between drug and contraction intensity for either high or low intensity stimulation (pg. 22, ln. 455 - 461). Does this not imply that the magnitude of units in the subliminal fringe is uniformly increased by cyproheptadine irrespective of contraction intensity, or are you attempting to normalize by motoneuron size? A greater emphasis on why the change metrics are critical for the subliminal fringe interpretation is needed, as it does not appear readily apparent.

We agree that the rationale for using normalised change scores should be more explicit to the readers and have updated the methods section (line 272) and discussion (line 623) accordingly.

While raw amplitudes provide insight into the proportion of motoneurons recruited by stimulation, the CMEP amplitudes increase with contraction intensity (from 10% MVC to 30% MVC). As such, a given change in absolute CMEP amplitude during stronger contractions may be equivalent to that observed as a large change during weaker contractions which may reflect a smaller relative effect. Indeed, there was no interaction effect between drug and contraction intensity for raw amplitudes. However, this could be due to potential masking effects during stronger contractions as a greater proportion of motoneurons are active and more neurons are inherently closer to firing thresholds. To account for the effect of contraction intensity on CMEP amplitude, relative change score metrics (% of control) were calculated. This change score metric was also used to account for between-individual variability in baseline CMEP amplitudes, and session-to-session variability, while allowing for direct within-subject comparisons between placebo and cyproheptadine conditions. Importantly, relative change scores allow us to directly examine the magnitude of change between low and high intensity stimulation across contraction intensities within each condition.

Nonetheless, for transparency of the data, both raw amplitudes and change scores are presented in the results and converging interpretations are drawn from both analyses. To address this concern raised by Referee #1, we have now added the following consideration to the manuscript, beginning at line 623:

- “While it could be interpreted that these CMEP findings suggest lower threshold motoneurons may be more affected by 5-HT₂ antagonism than higher threshold motoneurons, some consideration should be given to the application of relative change score metrics (% of control). While percentage change scores were employed to account for individual variability in baseline CMEP amplitude, small changes to raw amplitudes may reflect large percentage changes, particularly during low intensity contractions when CMEP amplitudes are small. Conversely, the alternative approach using absolute comparisons of change (post – pre) is highly dependent on baseline amplitudes which are larger during stronger contractions and does not account for changes in amplitude relative to contraction intensity. Therefore, interpretations of differential serotonergic effects on low and high threshold motoneurons remains speculative, as further exploration is needed for direct comparisons between low and high threshold motoneurons. Nevertheless, these CMEP findings provide insight into the effects of voluntary drive compensating for 5-HT₂ antagonism across different levels of motoneuron activation, probed by low and high intensity stimulation.”

Minor Comments/Considerations

Have the authors given any consideration to the potential histaminergic effects of cyproheptadine? Histamine does modulate motor behaviors across various animal models. Could histaminergic-mediated modulation play a role in any of the reported findings?

Yes, we had considered that the drug effects on histamine receptors when designing this experiment and interpreting the results. We have now added the following statements to the ‘*Considerations*’ section of the discussion.

- “While the present findings provide novel insights to the serotonergic modulation of the motoneurone pool in humans, some methodological considerations should be acknowledged. Firstly, cyproheptadine hydrochloride is a competitive antagonist of histamine H₁ receptors and 5-HT₂ receptors, whilst also exhibiting minor anticholinergic properties by acting on muscarinic receptors. Antagonism of H₁ and M₁ receptors have previously been investigated with strong antihistamines, where promethazine hydrochloride was found to have no effect on motoneurone excitability following muscle contraction (Dempsey & Kavanagh, 2021), motor evoked potential amplitude during muscle contraction (Dempsey & Kavanagh, 2023), or short latency inhibition or facilitation with the muscle at rest (Di Lazzaro et al., 2000). Although potential histaminic effects on motoneurone excitability from cyproheptadine ingestion are unlikely, the influence of these potential effects on the results of the current study cannot be completely ruled out.”

The word "indirectly" is used in the title, but only in the title. It is unclear what the authors intended to portray with this. While it is presumed that this indirect nature refers to the changes in necessary excitatory drive induced by PIC changes, the authors should better highlight what they mean by indirect effects in the text (introduction and/or discussion).

To clarify what is meant by the word “indirectly” in the title, the following text has been added to the key points and introduction:

- Key points (line 34): “These findings suggest that compensatory voluntary drive is required to achieve the same torque level with 5-HT₂ receptor antagonism, indirectly enhancing motoneurone activity and excitability of the motoneurone pool.”
- Introduction (line 93): “Hence, the increase in CMEP amplitude is likely attributable to an ‘indirect’ effect of drug ingestion; whereby, increased excitatory drive is required to achieve the same level of torque due to altered motoneurone excitability caused by 5-HT₂ antagonism.”

What was the rationale used for the dosage (8 mg) of cyproheptadine, as well as a static dose across all participants?

The 8mg dosage for cyproheptadine hydrochloride was chosen as it is double the therapeutic dose but remains within a safe administration amount without significantly increasing the risk of adverse effects. By selecting a dose of 8mg, there will be strong antagonistic effects on 5-HT₂ receptors. As such, this dosage has been used previously and found to exert strong effects on evoked potentials and motor unit firing characteristics (Goodlich et al., 2023; Henderson et al., 2023; Thorstensen et al., 2021, 2022). While individualised dosage for participants was considered when designing this study, a standardised dosage was deemed more appropriate for the experimental design. This is because individualised dosage based on

body weight does not control for differences in muscle/adipose mass or secretion rates of the drug between participants.

Were smooth or instantaneous estimates of discharge used for the quantification of mean discharge rate? The authors should clarify this (somewhere around pg. 11, ln. 292), since they show smoothed discharge rates in Figure 3A. Also, if smoothed discharge rates were used, were the hyperparameters suggested by Beauchamp et. al., used or something different?

Instantaneous discharge rates were used for the analysis of motor unit characteristics. Line 298 now reads: “Firing characteristics were calculated from instantaneous estimates of discharge for tracked units from pre-pill to post-pill ingestion for placebo and cyproheptadine testing sessions.” Smooth discharge rates were calculated using Support Vector Regressions produced by machine learning (support vector machine) as outlined by (Beauchamp et al., 2022), and used only as a visual representation of motor unit discharge characteristics for representative data in this figure.

How many units on average per participant were decomposed?

Following decomposition, an average of 39 and 32 units were identified for each participant across contraction intensities for placebo and cyproheptadine conditions, respectively.

Line 397 now reads, “Following decomposition, an average of 39 and 32 motor units were identified for each participant across contractions intensities for placebo and cyproheptadine conditions, respectively. Of these units identified, an average of 16 and 13 motor units were tracked for each participant from baseline to post-pill for placebo (176 total), and from baseline to post-pill for cyproheptadine (145 total) condition, respectively.”

Figure 3: Could you indicate matched units from pre- to post-pill ingestion with matching colors here? An increase in discharge rate is reported consistently in the results, but the example shown makes that hard to appreciate.

Yes, the representative figure (now figure 4) has colours associated with each individual motor unit tracked from pre- to post-pill.

Table 2 & Figure 4: It would be helpful if the EMM and LMM results from Figure 4 were shown quantitatively in Table 2. I know this is a personal preference, but it makes interpretation far easier if you can appreciate both the qualitative and quantitative aspects.

This data has now been added to Table 2.

Pg.24, ln. 485: "(p=0.0.587)"

This has now been changed to "(p = 0.587)"

Pg. 26, ln 514: If motor unit synchronization is thought responsible, couldn't you investigate this with coherence or metrics of shared synaptic input (Negro et. al.,).

Agreed, coherence metrics could provide interesting insights to whether motor unit synchronisation is contributing to these results. However, due to the low number of motor units tracked pre- and post-drug ingestion for cyproheptadine in the current study, this analysis would have low statistical power and would not accurately represent the population of the motoneurone pool. Several lines of evidence suggest many units provide higher accuracy of coherence estimation and low intensity contractions should be sustained for long durations to identify more motor units that are simultaneously active (Farina et al., 2014; McManus et al., 2019; Negro & Farina, 2012). Therefore, this claim remains speculative and will be explored in future investigations.

Pg. 29, ln. 600: Double period after amplitude

This typo has now been corrected.

Pg. 29, ln. 611: PICs being voltage-dependent does not necessitate voluntary drive for their activation. Input that drives the membrane suprathreshold long enough to allow activation of the PIC would likely be sufficient, irrespective of its voluntary or involuntary nature.

Agreed, this sentence has now been changed to the following "Due to their voltage-dependence, PICs are only active when motoneurons receive sustained input to raise their membrane potentials near threshold. During 10% contractions, this applies to the small proportion of the motoneurone pool voluntarily recruited"

Referee #2:

This paper builds on recent findings from the same research group, which demonstrated that 5-HT₂ antagonism increases cervicomedullary motor-evoked potentials (CMEPs) in the right biceps brachii. Expanding on these results, the present study investigates the effects of 5-HT₂ antagonism on biceps brachii motor neuron excitability by combining high-density surface EMG (HDsEMG) recordings with cervicomedullary stimulation during voluntary contractions. Overall, the study addresses an interesting and relevant research question and is well conducted, with a clearly identified research gap, well-defined purpose, and explicit hypotheses. However, several methodological and analytical concerns must be addressed to strengthen the conclusions. I outline these points below.

We thank Referee 2 for their comments and can appreciate the concerns outlined in the comments below.

One of the most critical concerns is the recording of CMEPs. What was the approximate inter-electrode distance of the bipolar EMG electrodes (Lines 178-179)?

The interelectrode distance for bipolar EMG was ~4 cm and has now been included in line 165.

It appears that the inter-electrode distance may have been relatively large. While a greater inter-electrode distance increases conduction volume and captures a more representative response of muscle excitation, it also raises the risk of crosstalk from adjacent muscles (e.g., brachialis), particularly during voluntary contractions. Did the authors assess potential crosstalk from other flexor muscles? This is crucial because while motor unit activity was recorded exclusively from the biceps brachii, there is a risk that CMEPs were recorded from both the biceps brachii and other muscles, which could compromise data interpretation. A straightforward way to address this concern would be to compare responses obtained from the bipolar EMG system with a bipolar configuration extracted from the electrode grid. If these responses are similar, it would confirm that CMEPs were exclusively recorded from the biceps brachii.

We acknowledge that despite an increased conduction volume and representative response, this increased inter-electrode distance can potentially increase the likelihood of crosstalk from adjacent muscles. Indeed, the HDsEMG grid overlaying the biceps brachii likely reflects activation of superficial motor units underlying the electrode grid, and the bipolar configuration may also include deeper units. Unfortunately, there were no measures of crosstalk during the elbow flexion tasks. As suggested, we performed an additional analysis on a subset of data, whereby a bipolar configuration from HDsEMG recordings was used to examine M_{MAX} and CMEP amplitudes from a similar interelectrode distance from the midpoint of the grid (electrode channels 31 and 36). Absolute amplitudes were larger in the HDsEMG configuration (as expected due to different amplification of signals between the two configurations, muscle recording locations, and electrode size etc), but similar in relation to M_{MAX} amplitude, and trial-to-trial variability and the relative change in amplitude were

highly consistent between both bipolar configurations. For example, in one contraction intensity, peak-to-peak amplitude was ~74% of M_{MAX} for the bipolar configuration and ~70% of M_{MAX} for the HDsEMG configuration, with a relative change of -1.2% and -1% respectively. This consistency between trials from evoked responses and between relative change calculations indicates similar physiological recordings of the motoneurone pool from both EMG configurations.

Given that the brachialis acts as a synergist to the biceps brachii during elbow flexion, it is likely that the effects of 5-HT₂ antagonism affects these muscles similarly. Thus, potential crosstalk from the brachialis muscle would not impact the interpretation of the results, and motor unit recordings and CMEP responses are treated separately in the statistical analyses and in the discussion. Nonetheless, phrases in the manuscript have been reframed to indicate HDsEMG is primarily reflective of biceps brachii motor unit activity, whereas CMEP amplitudes are referred to more broadly reflect activity of the elbow flexors. Notable changes are outlined below:

- Abstract (line 46): “Cervicomedullary stimulation was used to produce small and large CMEPs in the elbow flexors during these submaximal contractions.”
- Methods (line 198: “... single electrical stimuli at the cervicomedullary junction to elicit cervicomedullary motor evoked potentials (CMEPs) in the right elbow flexors.”
- Figure 2 caption: “High-density surface electromyography (HDsEMG) recordings were obtained from a 13x5 grid electrode overlaying the biceps brachii, and cervicomedullary stimulation was delivered at low (low stim) and high (high stim) stimulation intensities to produce cervicomedullary motor evoked potentials (CMEP) in the elbow flexors.”
- Conclusion (line 694): “This project found that motor unit discharge rates and CMEP amplitudes of the elbow flexors both increase with 5-HT₂ antagonism during submaximal elbow flexions when torque targets remain unchanged from baseline (pre-pill ingestion).”

Additional details regarding brachial plexus stimulation are needed. What were the dimensions of the cathode and anode?

The electrodes used for the cathode and anode were 24 mm. Line 82 now reads “Maximal compound muscle action potentials (M_{MAX}) were evoked in the biceps brachii using a constant current stimulator (0.2 ms pulse width, DS7AH, Digitimer Ltd., UK) and Ag/AgCl electrodes (Kendall ARBO, 24 mm diameter)”

How was the stimulation intensity for Mmax determination selected (e.g., staircase protocol)?

To determine M_{MAX} amplitude, a staircase protocol was followed and M_{MAX} was verified by delivering a minimum of five supramaximal stimulations (up to ~150% of the intensity required to produce the maximal M-wave). The following sentence has now been added to

line 185 “To determine M_{MAX} amplitude, the stimulation intensity was gradually increased in increments of 5-20 mA until a clear plateau was identified in the M-wave amplitude.”

A key concern is that the stimulation intensity remained unchanged across sessions. Surface electrode stimulation of the brachial plexus is highly sensitive to slight variations in electrode positioning and skin-electrode contact. Therefore, it is essential to report intra-session reliability measures (intraclass correlation coefficients) to ensure that M-wave amplitudes elicited at the same intensity were reproducible across sessions. Similarly, intra-session reliability should be reported for CMEP responses, as the same stimulation intensities ("low" and "high" stimulation) were used across sessions.

We acknowledge the concern with stimulation intensities remaining the same pre-pill to post-pill sessions within drug conditions and have reported ICC's in the revised manuscript for M_{MAX} . Line 190 now reads: “To confirm reliability of M-wave measures across testing sessions, intraclass correlation coefficients (ICC) were calculated for M_{MAX} amplitude between pre-pill and post-pill sessions in the placebo condition (10%: ICC = 0.98, 20%: ICC = 0.98, and 30% MVC: ICC = 0.97) and the cyproheptadine condition (10%: ICC = 0.95, 20%: ICC = 0.96, and 30% MVC: ICC = 0.97).”

However, it does not seem appropriate to perform this calculation on CMEP data. CMEP amplitudes were normalised to relative M_{MAX} for each contraction intensity within each testing session, and CMEP amplitude was expected to change with drug ingestion. Stimulation intensities remained fixed between sessions (i.e. pre pill to post pill on the same day) to ensure that similar levels of additional synaptic input were provided to the motoneuron pool post drug ingestion. Adjusting the level of electrical current would likely mask the effects of cyproheptadine as it would not be consistent between sessions and changes in amplitude may be attributable to changes in stimulation intensity. Furthermore, electrodes remained attached to the participant between sessions (taped securely to the skin), electrode contact was checked regularly, and participants were supervised at all times to ensure there were no problems with EMG recordings. The analysis that compares pre to post change scores for the drug and placebo days accounts for potential measurement variability between the pre- and post-pill sessions.

It is unclear why separate ANOVAs were used to compare the effects of placebo vs. cyproheptadine conditions and the effects of low vs. high-intensity stimulation on CMEPs. Why was stimulation intensity not included as a cofactor in the first analysis? A linear mixed-effects model, as used in the motor unit analysis, could also be applied to assess evoked responses, potentially providing a more comprehensive statistical approach.

We appreciate the Referee's request to perform similar LMM analyses on the CMEP data. Multiple two-way repeated measures ANOVA were used in the original analysis to simplify the readability and interpretation of the results. By performing a LMM with four fixed effects and their interactions (as suggested), the models become quite complex and the interpretations require multiple steps. For the Referee's interest, we have applied LMM's for the CMEP data and included these results in this Response Letter (Table 1 and 2 below). For

pre-pill to post-pill comparisons, there are fixed effects for testing session (pre-pill, post-pill), drug condition (placebo, cyproheptadine), contraction intensity, and stimulation intensity, as well as their interactions.

The LMM's used for CMEP analyses reflect the same differences observed using multiple ANOVAs. However, the interpretation of these results can be confusing as there is no significant main effect of drug condition for the pre- to post-pill comparisons. This is largely due to the baseline (pre-pill) session for cyproheptadine being included in the comparison to placebo, which may mask the effects of drug ingestion. Instead, a significant interaction effect and subsequent post hoc determines this difference, indicating that CMEP amplitudes obtained post-pill cyproheptadine are larger than pre-pill. Given that the same results are found using the suggested LMM approach, but requires a more complex interpretation, the two-way repeated measures ANOVA's have remained in the statistical analysis in the manuscript.

Table 1: Pre-post comparisons (linear mixed model)

Factors	F(df)	P-value
Testing session (pre- and post-pill)	20.7 (1, 207)	< 0.001*
Drug condition (placebo and cyproheptadine)	0.2 (1, 207)	0.679
Contraction intensity	112.0 (2, 207)	< 0.001*
Stimulation intensity	778.7 (1, 207)	< 0.001*
Drug condition by session (interaction)	5.3 (1, 207)	0.023*
Interaction: drug condition by stimulation intensity	6 (1, 207)	0.015*
Interaction: drug condition by contraction intensity	0.1 (1, 207)	0.968

Post hoc analysis (EMM comparisons)	T(df)	P-value
Placebo: pre to post	1.5 (233)	0.135
Cyproheptadine: pre to post	4.6 (233)	< 0.001*
Placebo: low stim vs high stim	20.2 (233)	< 0.001*
Cyproheptadine: low stim vs high stim	17.0 (233)	< 0.001*

Table 2: Change score comparisons (linear mixed model)

Factors	F(df)	P-value
Drug condition (placebo and cyproheptadine)	53.7 (1, 99)	< 0.001*
Contraction intensity	3.7 (2, 99)	0.028*
Stimulation intensity	5.6 (1, 99)	0.020*
Interaction: drug condition by stimulation intensity	3.7 (1, 99)	0.059
Interaction: drug condition by contraction intensity	1.35 (2, 99)	0.265

Post hoc analysis (EMM comparisons)	T(df)	P-value
Placebo: low stim vs high stim	0.3 (111)	0.767
Cyproheptadine: low stim vs high stim	2.9 (111)	0.005*
10% contraction: low stim vs high stim	2.0 (111)	0.046*
20% contraction: low stim vs high stim	1.1 (111)	0.3
30% contraction: low stim vs high stim	0.8 (111)	0.43

The rationale behind selecting 15%, 25%, and 35% MVC for calibration contractions (motor unit filter extraction) while using different torque levels (10%, 20%, and 30% MVC) for ramped contractions is unclear. Please provide an explanation for this discrepancy.

The following explanation has now been added to the experimental protocol section of the Methods, Line 224 “The additional calibration contractions of 15%, 25% and 35% were used to train the motor unit filters at six different torque levels (10%, 15%, 20%, 25%, 30% and 35%) rather than three torque targets. The reason for this was to improve the sensitivity of motor unit decomposition across a wider range of recruitment thresholds. A secondary purpose of this was to increase the likelihood of identifying additional higher threshold units that may be recruited with cyproheptadine as a result of increased descending drive.”

While a sample size of approximately ten participants is common in this field, it remains relatively small. Please provide a sample size calculation to justify the adequacy of the chosen sample. Additionally, there is no mention of whether written informed consent was obtained from participants. Please ensure this information is included.

Written informed consent was obtained from all participants prior to participation in this study. Line 141 now reads “and all participants provided written informed consent prior to any testing procedures.”

Given the placebo-controlled, within-subject crossover design of this study, a sample size of ~10 participants was deemed suitable to achieve statistical power. A power calculation using G*Power was completed prior to participant recruitment using a modest effect size (0.45) achieved for measures of evoked potentials under similar conditions from previous studies (Henderson et al., 2023; Thorstensen et al., 2022). This calculation (figure below) indicated statistical power of ~80% can be achieved with 10 participants. While it is acknowledged that 10 participants may be relatively small for some studies in this field, the robust experimental design, requirement for drug ingestion of a strong anti-serotonergic, high intensity cervicomedullary stimulation, and time commitment of participants, meant that we did not want to unnecessarily recruit participants if statistical power is already achieved.

To improve clarity, please include schematic figures illustrating electrode placement for both stimulation and recording. This would help readers better understand electrode positioning, particularly for cervicomedullary stimulation.

The following figure has now been included in the participant setup section the methods.

Participant setup. Participants were seated upright, with their right arm attached to a custom torque transducer. Elbow flexion torque was measured with the elbow and shoulder joints flexed at 90 degrees. Bipolar EMG and a 64 channel (5 columns by 13 rows) high-density EMG array were positioned over the biceps brachii muscle. Electrodes for electrical stimulation were positioned over the brachial plexus (Erb's point) and slightly inferior and medial to the left and right mastoid process.

Minor comments:

Line 93: Please introduce the term "CMEP" in full before using the abbreviation.

This line has been corrected to the following "...found to increase cervicomedullary motor evoked potential (CMEP) amplitude..."

Lines 140-141: Please include a reference to support this statement.

The following paper by Paton and Webster (1985) titled "Clinical pharmacokinetics of H1-receptor antagonists (the antihistamines)" reports the half-life of cyproheptadine as 16 hours. After 7 half-lives of the drug (~5 days), there will be less than 1% of the drug remaining in the system. This reference has now been added to the statement beginning on line 140.

Lines 152-153: Provide additional details on participant characteristics, including height and weight.

Participant height (178 ± 8 cm) and weight (80 ± 10 kg) have now been added to this line in the manuscript.

Lines 178-179: How were the midpoint of the muscle belly and the distal tendon of the biceps identified? Was ultrasound or palpation used? Please clarify.

The midpoint of the muscle belly and the distal tendon were identified by palpation. The following sentence has now been added “These locations were identified by palpation by an experienced investigator.”

Lines 187-189: Please specify the gain settings for the HDsEMG signals.

The sentence has now been updated to include the amplification of the signal “Monopolar HDsEMG signals were amplified (x 256), processed and sampled at 2000 Hz using a wireless amplifier.”

Lines 260-262: The description of "initial deflection in the bipolar EMG signal to the second crossing of 0 mV" is unclear. Please provide further details for clarity.

To provide a clearer description, line 260 now reads “ M_{MAX} and CMEP responses were measured as the peak-to-peak amplitude of the biphasic waveform for each contraction. This was conducted by setting a horizontal cursor at 0-mV, and the evoked responses were measured from the first clear deflection of the EMG signal following the stimulus artefact, to where the EMG signal returns to the 0-mV line following both phases of the waveform.”

Lines 398-399: Include the average number of tracked motor units per participant.

An average of 16 and 13 units were tracked for participants on placebo and cyproheptadine testing days respectively. Line 398 now reads, “Following decomposition, an average of 39 and 32 motor units were identified for each participant across contractions intensities for placebo and cyproheptadine conditions, respectively. Of these units identified, an average of 16 and 13 motor units were tracked for each participant from baseline to post-pill for placebo (176 total), and from baseline to post-pill for cyproheptadine (145 total) condition, respectively.”

Dear Dr Henderson,

Re: JP-RP-2025-288317R1 "5-HT2 antagonism indirectly increases motor unit discharge rate and cervicomedullary motor evoked potential amplitude during submaximal elbow flexions" by Tyler T Henderson, Janet L Taylor, Jacob Thorstensen, and Justin J Kavanagh

Thank you for submitting your manuscript to The Journal of Physiology. It has been assessed by a Reviewing Editor and by 2 expert referees and we are pleased to tell you that it is acceptable for publication following satisfactory revision.

REVISION CHECKLIST:

We look forward to receiving your revised submission.

Yours sincerely,

Richard Carson
Senior Editor
The Journal of Physiology

EDITOR COMMENTS

Reviewing Editor:

The authors have carefully considered and addressed the comments of referees. There are still a few outstanding suggestions from reviewers that the authors should consider to facilitate clarity and transparency.

REFEREE COMMENTS

Referee #1:

The authors are recognized for their comprehensive response to the reviewers' comments. I have no further critical concerns that would affect the suitability for publication.

Referee #2:

I thank the authors for their thoughtful responses to the previous round of reviews. All of my earlier comments have been addressed satisfactorily, and the manuscript has been substantially improved. I have only a few minor suggestions that I believe will further strengthen the clarity and reproducibility of the work:

1) I appreciate the authors' adjustment in terminology to use the broader term "elbow flexors" when referring to CMEP amplitudes. However, I still recommend reporting that similar physiological responses were obtained when using a bipolar configuration extracted from the electrode grid. Although discussed separately, the CMEP and motor unit results are ultimately used together to support the study's interpretations. Therefore, it is important to confirm that the CMEP responses, when analyzed specifically from the biceps, yielded consistent results.

2) Lines 194-198: Please report the 95% confidence intervals for the ICC values and include the specific ICC model used to improve reproducibility.

3) Please add the description of the sample size calculation in the manuscript (not only in the reply), including the assumptions made (e.g., effect size, power, alpha level).

4) Lines 398-399: Please report a measure of dispersion (e.g., standard deviation) alongside the average number of tracked motor units.

END OF COMMENTS

EDITOR COMMENTS

Reviewing Editor:

The authors have carefully considered and addressed the comments of referees. There are still a few outstanding suggestions from reviewers that the authors should consider to facilitate clarity and transparency.

We thank both Referees for their constructive reviews. We have now added all minor suggestions raised by Referee #2 to the revised manuscript and feel this has improved the clarity and transparency of the study.

REFEREE COMMENTS

Referee #1:

The authors are recognized for their comprehensive response to the reviewers' comments. I have no further critical concerns that would affect the suitability for publication.

We appreciate the kind response and consideration of this manuscript.

Referee #2:

I thank the authors for their thoughtful responses to the previous round of reviews. All of my earlier comments have been addressed satisfactorily, and the manuscript has been substantially improved. I have only a few minor suggestions that I believe will further strengthen the clarity and reproducibility of the work:

1) I appreciate the authors' adjustment in terminology to use the broader term "elbow flexors" when referring to CMEP amplitudes. However, I still recommend reporting that similar physiological responses were obtained when using a bipolar configuration extracted from the electrode grid. Although discussed separately, the CMEP and motor unit results are ultimately used together to support the study's interpretations. Therefore, it is important to confirm that the CMEP responses, when analyzed specifically from the biceps, yielded consistent results.

This has now been added to the methods section. Line 287 now reads: "To confirm that evoked potentials were indeed obtained from the biceps brachii muscle, M_{MAX} and CMEP amplitudes from a subset of participants were also analysed using a bipolar configuration from the HDsEMG electrode recordings. Whereby, similar physiological responses were obtained from both forms of bipolar EMG analyses and yielded consistent results. For

example, in one contraction intensity, peak-to-peak amplitude was ~74% of M_{MAX} for the bipolar configuration and ~70% of M_{MAX} for the HDsEMG configuration, with a relative change of -1.2% and -1% respectively.”

2) Lines 194-198: Please report the 95% confidence intervals for the ICC values and include the specific ICC model used to improve reproducibility.

To confirm reliability of M-wave measures across testing sessions, intraclass correlation coefficients (ICC) were calculated for M_{MAX} amplitude between pre-pill and post-pill sessions with two-way mixed effects models using the psych package, version 2.5.3 (Revelle., 2025), in R Studio. These models revealed excellent reliability between sessions in the placebo condition (10%: ICC = 0.98, 95% CI = 0.93, 0.99, 20%: ICC = 0.98, 95% CI = 0.90, 0.99, and 30% MVC: ICC = 0.97, 95% CI = 0.87, 0.99) and the cyproheptadine condition (10%: ICC = 0.95, 95% CI = 0.83, 0.99, 20%: ICC = 0.96, 95% CI = 0.82, 0.99, and 30% MVC: ICC = 0.97, 95% CI = 0.86, 0.99).

3) Please add the description of the sample size calculation in the manuscript (not only in the reply), including the assumptions made (e.g., effect size, power, alpha level).

Line 129 now reads “Prior to participant recruitment, a sample size calculation was completed to estimate the number of required participants to achieve statistical power. An effect size (0.45) observed from measures of evoked potentials under similar conditions in our previous work (Henderson et al., 2023; Thorstensen et al., 2022), an alpha level 0.05, and power of 0.8 was used in this calculation, where an estimated 10 participants was required for this experimental design. Considering possible experimental challenges (e.g., discomfort from electrical stimulation) and potential participant drop-out, fourteen healthy individuals were recruited to participate in this study.”

4) Lines 398-399: Please report a measure of dispersion (e.g., standard deviation) alongside the average number of tracked motor units.

The standard deviation has now been added for the average number of motor units decomposed (39 ± 16 and 32 ± 16) and motor units tracked between sessions for each condition (16 ± 6 and 13 ± 7).

Dear Dr Henderson,

Re: JP-RP-2025-288317R2 "5-HT2 antagonism indirectly increases motor unit discharge rate and cervicomedullary motor evoked potential amplitude during submaximal elbow flexions" by Tyler T Henderson, Janet L Taylor, Jacob Thorstensen, and Justin J Kavanagh

We are pleased to tell you that your paper has been accepted for publication in The Journal of Physiology.

Yours sincerely,

Richard Carson
Senior Editor
The Journal of Physiology

If you would like to receive our 'Research Roundup', a monthly newsletter highlighting the cutting-edge research published in The Physiological Society's family of journals (The Journal of Physiology, Experimental Physiology, Physiological Reports, The Journal of Nutritional Physiology and The Journal of Precision Medicine: Health and Disease), please click this link, fill in your name and email address and select 'Research Roundup':
<https://www.physoc.org/journals-and-media/membernews>

- You can help your research get the attention it deserves! Check out Wiley's free Promotion Guide for best-practice recommendations for promoting your work at: www.wileyauthors.com/eeo/guide. You can learn more about Wiley Editing Services which offers professional video, design, and writing services to create shareable video abstracts, infographics, conference posters, lay summaries, and research news stories for your research at: www.wileyauthors.com/eeo/promotion.

EDITOR COMMENTS

Reviewing Editor:

Thank you to the authors for a quick response. There are no outstanding comments or concerns.